# Adaptive Preference Arithmetic: Modeling Dynamic Preference Strengths for LLM Agent Personalization

**Hongyi Nie**
Northwestern Polytechnical
University
hy_nie@mail.nwpu.edu.cn

**Yaqing Wang**[*]
Beijing Institute of Mathematical
Sciences and Applications
wangyaqing@bimsa.cn

**Mingyang Zhou**
Tsinghua University
zhou-my24@mails.tsinghua.edu.cn

**Feiyang Pan**
Tsinghua University
pfy824@gmail.com

**Quanming Yao**
Tsinghua University
yaoaa@tsinghua.edu.cn

**Zhen Wang**[*]
Northwestern Polytechnical
University
w-zhen@nwpu.edu.cn

## Abstract

As large language models (LLMs) are increasingly used as personalized user assistants, effectively adapting to users' evolving preferences is critical for delivering high-quality personalized responses. While user preferences are often stable in content, their relative strengths shift over time due to changing goals and contexts. Therefore, *modeling these dynamic preference strengths can enable finer-grained personalization*. However, current methods face two major challenges: (i) limited user feedback makes it difficult to estimate preference strengths accurately, and (ii) natural language ambiguity limits the controllability of preference-guided generation. To address these issues, we propose **AdaPA-Agent**, a LLM-agent personalization framework that models dynamic preference strengths via *Adaptive Preference Arithmetic*. First, instead of requiring additional user feedback, AdaPA-Agent employs an alignment-based strength estimation module to estimate the strength of user preferences from the existing user-agent interaction. Then, it guides controllable personalized generation by linearly combining next-token distributions, weighted by the estimated strengths of individual preferences. Experiments on two personalization tasks-conversational recommendation and personalized web interaction-demonstrate that AdaPA-Agent better aligning with users' changing intents, and has achieved over 18.9% and 14.2% improvements compared to ReAct, the widely-used agent framework.

## 1 Introduction

Agents powered by large language models (LLMs) [1, 2] are increasingly being utilized as user assistants [3], helping individuals with tasks such as information retrieval and decision-making. Given that user needs are highly diverse and personalized, they often do not want assistants to provide generic responses. Instead, users expect these user assistants to understand their unique needs better and provide personalized services [4], such as travel planning [5], and shopping recommendations

---

[*]Corresponding authors

[6, 7]. Therefore, enhancing the ability of LLM agents to model users' personalized needs has become a key focus of current research [3].

Since user preferences fundamentally shape their personalized needs, recent approaches improve personalization of LLM agents by modeling and leveraging these preferences [8]. These methods can be divided into two categories: fine-tuning and training-free methods. Fine-tuning [9, 10] involves adjusting the model's parameters to fit personal preferences. However, this approach requires gathering large amounts of individual data and model training, making it both resource-intensive and costly. In contrast, training-free methods such as retrieval-augmented generation (RAG) and prompt engineering provide more flexible alternatives. RAG [8, 11] enables agents to retrieve relevant preference information in real-time, adapting responses without retraining the model. Prompt engineering [12, 13, 14] inserts the prompts of preference content into prompt templates, which can guide the model toward personalized outputs without modifying its underlying parameters.

In real-world scenarios, users often hold multiple coexisting preferences–such as favoring both healthy meals and junk food–but the influence of each preference on user's decision-making is not equal and can vary over time [15]. We term this influence as *preference strength*. As shown in Figure 1, without modeling this dynamic variation in preference strength, LLM agents may produce responses that fail to align with the user's current intent.

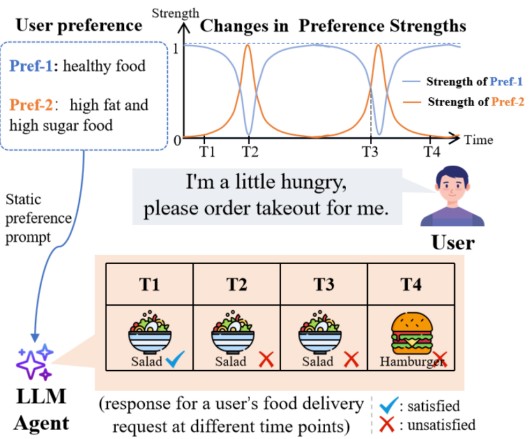

Although, existing methods make great progress in personalization, they still face two key challenges of modeling the dynamic preference strengths for LLM agents: (i) *Accurately estimating preference strengths without relying on explicit user feedback*. Existing methods such as ReAct [16] and Reflexion [17] often require user feedback to help adjust the strengths of preferences. Yet in real-world scenarios, user feedback is often limited in number, and frequent requests for feedback will disrupt normal interactions and significantly degrade user experience. (ii) *Effectively utilizing estimated strengths to guide content generation*. A common strategy is to input the preference strengths as text prompts to LLMs. However, this prompt-based method is fundamentally limited by the inherent ambiguity of natural language [18, 19], which may lead to misaligned personalized responses.

Figure 1: Illustration of an agent responding to a user's food delivery request over time. The user's needs are shaped by two preferences (Pref-1 and Pref-2), whose relative importance changes. To respond effectively, the agent must recognize and adapt to these shifts; otherwise, it may fail to meet expectations.

To address these challenges, we propose AdaPA-Agent, a framework that models and applies dynamic preference strengths through *Adaptive Preference Arithmetic*. AdaPA-Agent comprises two key components. First, to estimate preference strengths without explicit user feedback (Challenge (i)), we introduce **Alignment-Based Strength Estimation**, which combines *dual-side augmentation* with an *LLM-based alignment scorer* to measure how well each preference aligns with the current user-agent interaction. Second, to utilize these estimated strengths in generation (Challenge (ii)), we propose **Controllable Personalized Generation**. This component modulates the output distribution of an LLM by *linearly combining next-token distributions* conditioned on individual preferences and weighted by their inferred strengths. We validate AdaPA-Agent on two personalized agent tasks: *conversational recommendation* and *personalized web interaction*. Experimental results show that AdaPA-Agent consistently improves personalization quality over strong baselines, enabling LLM agents to generate responses that better reflect users' evolving intents and priorities.

Our key contributions can be summarized as follows:

- We identify two key challenges of modeling dynamic preference strengths for LLM-agent personalization: (i) accurately estimating preference strengths without relying on explicit user feedback, and (ii) effectively utilizing these estimated strengths to guide content generation.

- To address these challenges, we propose **AdaPA-Agent** that models dynamic preference strengths for LLM agents through *Adaptive Preference Arithmetic*. AdaPA-Agent introduces two novel components: *Alignment-Based Strength Estimation* for estimating preference strengths without explicit feedback, and *Controllable Personalized Generation*, which controls the personalized generation process by linearly combining next-token distributions of LLM based on the estimated strengths.

- We validate AdaPA-Agent on conversational recommendation and personalized web interaction tasks, achieving over 18.9% and 14.2% improvements compared to widely-used agent framework ReAct, respectively, demonstrating its effectiveness in adapting to users' dynamic preferences.

## 2 Related Work

### 2.1 LLM Agent

The emergence of LLMs has transformed autonomous agents, enabling them to perform tasks with human-like intelligence through powerful reasoning and knowledge capabilities. These agents typically feature modules for profiling [4], memory [20], planning [21], and action [22]. Profiling defines the agents role and personality [23, 24], while memory enables learning and adaptation [20]. Planning is enhanced by Chain-of-Thought (CoT) [25] and Tree of Thoughts (ToT) [26], which break down complex tasks using sequential or tree-based reasoning. Action modules execute plans by combining internal logic with external tools [27, 22]. Integrated reasoning and acting approaches, like ReAct [16], enable dynamic decision-making, while Reflexion [17] improves adaptability via self-reflection and feedback.

LLM agents are now widely used in domains like marketing [28], and software development [29]. As these agents become more capable, user demands have shifted from seeking general responses to expecting interactions that are more aligned with their personalized needs. Enhancing the ability of agents to provide tailored and personalized experiences has thus become a key focus of current research.

### 2.2 LLM Personalization

Real-world applications like personalized web interaction [27] and conversational recommendation [30] demand flexible models that adapt to evolving user needs. While traditional methods like collaborative filtering [31] are common, they lack semantic understanding and rich interactions. With LLMs, there's a shift toward language-based agents, offering more personalized and responsive experiences. LLM personalization mainly follows two approaches: fine-tuning and training-free methods. Fine-tuning [9, 10, 32] adjusts model parameters using user-specific data for improved alignment, though it requires significant resources. Training-free approaches like RAG and prompt engineering offer lightweight alternatives. RAG [8, 11] retrieves external data in real time, while prompt engineering [12, 13, 14, 33] adapts prompts to personalize outputs without modifying the model. Recent works further explore controllable text generation guided by user preferences. For instance, OPAD [34] and CoS [35] propose controllable generation methods that rely on explicit preference instructions to guide LLM outputs. AMPLe [36] infers preferences through multiple rounds of explicit user feedback.

However, current methods still struggle with modeling preference strengths and controlling generation when user feedback is sparse or inconsistent. Our work addresses these gaps by improving preference modeling and guiding LLM agents to produce more personalized outputs.

## 3 Target Formulation

This paper focuses on personalization tasks where LLM-based agents generate responses tailored to user preferences. This includes (i) *conversational recommendation*, where the agent interacts with users to refine intent and recommend suitable items, and (ii) *personalized web interaction*, where the agent performs context-aware actions such as search or comment generation based on inferred preferences. Here, we formulate the personalization task of LLM agent as follows:

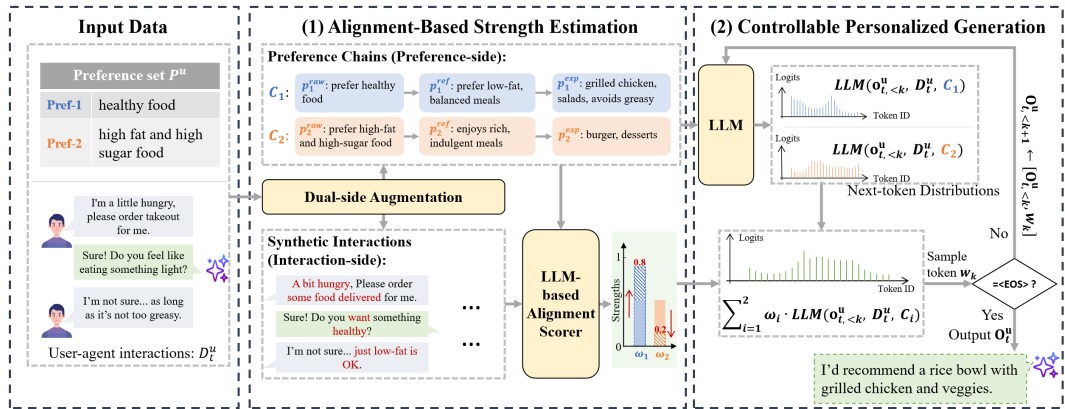

Figure 2: Overall framework of AdaPA-Agent, which includes two components: (1) **Alignment-Based Strength Estimation**, which computes alignment scores between structured preference chains and augmented user-agent interactions to infer the relative importance of preferences. (2) **Controllable Personalized Generation**, which combines next-token distributions from preference-conditioned LLM outputs, weighted by their estimated strengths. Our method enables the agent to generate responses that reflect the user's evolving priorities without requiring explicit feedback.

**Definition 3.1** *Personalization Task of LLM Agent [27, 6]. Given a user $u$, his user-agent interaction and intent at current time $t$ is $\mathcal{D}_t^u$ and $\mathcal{I}_t^u$, and historical interactions is $\mathcal{H}^u = \{\mathcal{D}_i^u\}_{i=1}^{t-1}$. The intent $\mathcal{I}_t^u$ is influenced by the user's preference set $\mathcal{P}^u = \{p_i^u\}_{i=1}^M$, where $M$ is the number of preferences. $\mathcal{P}^u$ can be extracted from $\mathcal{H}^u$, and each preference $p_i^u$ has a corresponding strength $\omega_i$. The goal of the LLM agent is to infer the preference information from user-agent interactions and use it to generate a response $\mathcal{O}_t^u$ that aligns with the user's intent $\mathcal{I}_t^u$.*

# 4 Methodology

Existing personalization methods often encode user preferences as static prompts or memories, overlooking how their *relative strength* evolves over time. However, modeling dynamic preference strength is challenging due to limited user feedback and the ambiguity of natural language. To address this, we propose **AdaPA-Agent**, a LLM-agent personalization framework that captures preference dynamics via *Adaptive Preference Arithmetic*. As shown in Figure 2, it includes: (1) **Alignment-Based Strength Estimation** (Section 4.1), and (2) **Controllable Personalized Generation** (Section 4.2). The first component estimates preference strengths adaptively without extra feedback through a dual-side augmentation module and an LLM-based alignment scorer. These strengths are then used in generation by combining next-token distributions conditioned on preferences in the second component. This allows the agent to adapt responses continuously based on updated preference weights. The following sections detail each component.

## 4.1 Alignment-Based Strength Estimation

Finding an observed signal that correlate the strength of user preferences is important for estimation without user feedback. Our key insight is that *the stronger a preference, the more frequently it manifests in user-agent interactions*. Based on the insight, we reformulate strength estimation as an *alignment* problem: for each preference, we measure how well it semantically aligns with the user-agent interaction and treat this alignment score as a proxy for strength. A higher alignment score indicates that the interaction provides stronger support for the preference, hence implying a larger strength for that preference.

However, user expressions are often vague and diverse in practice, which makes alignment scoring unreliable when using naïve methods such as keyword overlap or static embedding similarity. To address this challenge, we incorporate two key techniques: (i) dual-side augmentation, which enriches both the preference (via preference chains) and interaction sides (via conversation-level paraphrases) with more informative and semantically diverse representations; and (ii) an LLM-based

scoring module, which assesses the alignment between each augmented preference-interaction pair in a context-aware and interpretable manner.

### 4.1.1 Dual-Side Augmentation

**Preference-side augmentation.** We first extract the set of preferences $\mathcal{P}^u$ from historical interactions $\mathcal{H}^u$. However, user preferences often emerge at multiple semantic levels, i.e., from abstract self-descriptions to concrete behavioral cues. To capture this spectrum, we construct a three-stage *preference chain* for each preference $p_i^u$ using a CoT prompt:

$$\mathcal{C}_i \;=\; \mathrm{CoT}(p_i^u) \;=\; \left[ p_i^{\mathrm{raw}} \;\to\; p_i^{\mathrm{ref}} \;\to\; p_i^{\mathrm{exp}} \right], \tag{1}$$

where $p_i^{\mathrm{raw}}$ is the original description of the preference extracted from $p_i^u$, $p_i^{\mathrm{ref}}$ is a context-aware reformulation refining $p_i^{\mathrm{raw}}$, and $p_i^{\mathrm{exp}}$ enumerates concrete items or behaviors exemplifying $p_i^{\mathrm{raw}}$ and $p_i^{\mathrm{ref}}$. This three-level structure supplies coarse-to-fine preference information for the subsequent alignment scoring. The complete prompt is provided in Appendix A, Prompt 1.

**Interaction-side augmentation.** Since user intent is typically implicit and linguistically diverse, computing an alignment score from a single interaction can be highly biased. To reduce this variance, we enrich the interaction side by paraphrasing the original user-agent interaction $\mathcal{D}_t^u$ into multiple semantically equivalent variants:

$$\mathbf{g}(\mathcal{D}_t^u) \;=\; \left\{ \mathcal{D}_{t_1}^u, \, \mathcal{D}_{t_2}^u, \, \ldots, \, \mathcal{D}_{t_K}^u \right\}, \tag{2}$$

where we employ an LLM as a generation function $\mathbf{g}(\cdot)$ (Prompt 2, Appendix A), each synthetic interaction $\mathcal{D}_{t_k}^u$ ($k = 1, 2, \ldots, K$) preserves the same number of turns as $\mathcal{D}_t^u$ but rewrites utterances with alternative lexical and syntactic choices to enrich the expression of user intent. The augmented set $\mathcal{A}^u = \mathbf{g}(\mathcal{D}_t^u) \cup \mathcal{D}_t^u$ forms a more complete pool of interaction variants for alignment scoring.

### 4.1.2 LLM-Based Alignment Scoring

Compared to static embeddings or keyword matching, LLMs offer a richer understanding of semantics, enabling context-aware and fine-grained alignment. Therefore, we employ an LLM to assess the alignment score of each preference-interaction pair. Specifically, given the structured preference chains $\mathcal{C}_i$ and the augmented interaction set $\mathcal{A}^u$, we compute a alignment score $s_i$ for each preference $p_i^u$ as:

$$s_i \;=\; \frac{1}{|\mathcal{A}^u|} \sum_{\mathcal{D} \in \mathcal{A}^u} \mathbf{f}(\mathcal{C}_i, \mathcal{D}), \tag{3}$$

where $\mathbf{f}(\cdot, \cdot)$ is an LLM-based scorer returning a fine-grained alignment score (1-10) for a preference-interaction pair (Prompt 3, Appendix A). We then normalize the scores to obtain relative strength of each preference:

$$\omega_i \;=\; \frac{s_i}{\sum_{j=1}^M s_j}. \tag{4}$$

Through this process, we can obtain and adaptively update relative strength of each preference without requiring additional user feedback.

## 4.2 Controllable Personalized Generation

Once the set of preference strengths $\{\omega_i\}_{i=1}^M$ is obtained, the remaining question is how to use these weights to steer an LLM so that its response continuously reflects the user's current intent. Given the inherent ambiguity in natural language prompting, directly inserting numeric weights into a text prompt is unreliable [18]. Recent work [19] has demonstrated that complex textual style can be effectively generated by arithmetically combining the next-token distributions of language models conditioned on different basic styles. This motivates the proposal of preference arithmetic, a technique that combines multiple preference-conditioned distributions in a formulaic way, using the dynamically estimated strengths $\omega_i$ as weights. This allows for fine-grained control over the influence of each preference $p_i^u$ on the generated response $\mathcal{O}_t^u$.

Specifically, at each step of generating the response $\mathcal{O}_t^u$, the LLM produces a next-token distribution. Let $\mathbf{P}_{\mathrm{opt}}(\cdot)$ be the optimal personalized next-token distribution, which is conditioned on the

user's intent $\mathcal{I}_t^u$. We model $\mathbf{P}_{\text{opt}}(\cdot)$ as a weighted combination of $M$ individual next-token distributions, where each individual distribution $\mathbf{P}_{p_i^u}(\cdot)$ is conditioned on a specific preference $p_i^u$ from the user's preference set $\mathcal{P}^u$. The overall next-token distribution for generating the current token $w_k$ of the response $\mathcal{O}_t^u$, given the partially generated response prefix $\mathcal{O}_{t,<k}^u$ and the current user-agent interaction $\mathcal{D}_t^u$, is defined as:

$$\mathbf{P}_{\text{opt}}(w_k|\mathcal{O}_{t,<k}^u, \mathcal{D}_t^u, \mathcal{I}_t^u) = \sum_{i=1}^{M} \omega_i \cdot \mathbf{P}_{p_i^u}(w_k|\mathcal{O}_{t,<k}^u, \mathcal{D}_t^u, p_i^u), \tag{5}$$

where each preference-conditioned distribution $\mathbf{P}_{p_i^u}(\cdot)$ is generated by the LLM when prompted with the specific preference $p_i^u$ in the context of the current interaction $\mathcal{D}_t^u$ and the already generated part of the response $\mathcal{O}_{t,<k}^u$. For preference $p_i^u$, we can use its augmented representation, the preference chain $\mathcal{C}_i$ (from Section 4.1.1), to construct a comprehensive prompt for the LLM. Thus, the $i$-th preference-conditioned next-token distribution is:

$$\mathbf{P}_{p_i^u}(w_k|\mathcal{O}_{t,<k}^u, \mathcal{D}_t^u, p_i^u) = \text{LLM}((\mathcal{O}_{t,<k}^u, \mathcal{D}_t^u, \mathcal{C}_i)). \tag{6}$$

This formulation enables a continuously steerable decoding process, where each preference influences generation proportionally to its weight. Equivalently, the combined distribution can be viewed as the solution to a weighted KL minimization problem with closed-form optimum [19]:

$$P^\star = \arg\min_P \sum_{i=1}^{M} \omega_i \, D_{\text{KL}}\big(P \,\|\, Q_i\big), \tag{7}$$

$$P^\star(w_k) = \text{softmax}\Big(\sum_{i=1}^{M} \omega_i \log Q_i(w_k)\Big). \tag{8}$$

Here, $P^\star(w_k)$ represents the **KL-optimal compromise** among the preference-conditioned distributions $Q_i$, where stronger preferences (larger $\omega_i$) dominate while weaker yet relevant ones still contribute to a balanced, intent-aligned generation.

To generate the full response $\mathcal{O}_t^u$, tokens are sampled autoregressively. As illustrated in Algorithm 1, at each generation step $k$, the corresponding token $w_k$ is sampled from the combined distribution: $w_k \sim \sum_{i=1}^{M} \omega_i \cdot \text{LLM}((\mathcal{O}_{t,<k}^u, \mathcal{D}_t^u, \mathcal{C}_i))$, and extend the context as: $\mathcal{O}_{t,<k+1}^u = [\mathcal{O}_{t,<k}^u, w_k]$. This process continues iteratively until the sampled token $w_k$ is the end-of-sequence token (EOS), ensuring the generation of coherent and personalized responses.

## 5 Experiments

In this section, we evaluate our AdaPA-Agent[2] on two typical LLM personalized agent tasks (conversational recommendation and personalized web interaction). Additionally, we validate the effectiveness of our method design through a series of ablation experiments. In the following, we first present the experimental setup and baselines, and then provide a detailed presentation and analysis of the experimental results.

### 5.1 Experimental Setup

#### 5.1.1 Baseline

The proposed AdaPA-Agent is compared with the following baselines: **ReAct** [16] combines reasoning and action generation to improve decision-making in interactive tasks, allowing models to adapt dynamically to new information and user needs. **Reflexion** [17] uses verbal reinforcement learning for self-reflection and feedback, allowing language agents to continuously refine their actions and reasoning, making them more adaptable to personalized interactions. **SimToM** [37] introduces perspective-taking to improve theory-of-mind (ToM) capabilities, helping models better understand and predict human intentions, thus tailoring responses based on individual user preferences. **RecMind** [38] leverages an LLM-powered autonomous recommender agent with a self-inspiring algorithm, enabling zero-shot personalized recommendations by considering historical information and

---

[2]The code is available at: `https://github.com/Sirius11311/AdaPA`.

| Maximum Steps | Method | Long-Term Tasks | | Short-Term Tasks | | All Tasks | |
|---|---|---|---|---|---|---|---|
| | | RSR | AIR | RSR | AIR | RSR | AIR |
| 3 | ReAct [16] | $30.27_{\pm9.78}$ | $2.81_{\pm0.09}$ | $30.19_{\pm7.73}$ | $2.89_{\pm0.04}$ | $30.20_{\pm5.20}$ | $2.85_{\pm0.05}$ |
| | Reflexion [17] | $38.45_{\pm9.62}$ | $2.74_{\pm0.09}$ | $28.69_{\pm3.79}$ | $2.85_{\pm0.03}$ | $33.80_{\pm6.80}$ | $2.79_{\pm0.05}$ |
| | RecMind [38] | $39.77_{\pm9.65}$ | $\underline{2.73}_{\pm0.13}$ | $32.50_{\pm4.44}$ | $2.85_{\pm0.05}$ | $36.40_{\pm4.73}$ | $2.78_{\pm0.08}$ |
| | InteRec [39] | $\underline{45.67}_{\pm11.55}$ | $\mathbf{2.66}_{\pm0.16}$ | $\underline{32.87}_{\pm4.48}$ | $\underline{2.76}_{\pm0.08}$ | $\underline{39.60}_{\pm6.73}$ | $\underline{2.71}_{\pm0.10}$ |
| | SimTom [37] | $40.22_{\pm10.68}$ | $2.78_{\pm0.09}$ | $31.19_{\pm5.58}$ | $2.87_{\pm0.05}$ | $36.01_{\pm4.67}$ | $2.82_{\pm0.06}$ |
| | AdaPA-Agent | $\mathbf{46.10}_{\pm9.65}$ | $\mathbf{2.66}_{\pm0.10}$ | $\mathbf{37.05}_{\pm3.94}$ | $\mathbf{2.62}_{\pm0.06}$ | $\mathbf{41.45}_{\pm5.82}$ | $\mathbf{2.64}_{\pm0.08}$ |
| 5 | ReAct [16] | $57.47_{\pm9.53}$ | $4.19_{\pm0.19}$ | $40.99_{\pm4.59}$ | $4.46_{\pm0.09}$ | $49.60_{\pm4.60}$ | $4.32_{\pm0.14}$ |
| | Reflexion [17] | $67.88_{\pm7.80}$ | $3.81_{\pm0.24}$ | $51.40_{\pm6.30}$ | $4.24_{\pm0.18}$ | $60.20_{\pm4.80}$ | $4.01_{\pm0.19}$ |
| | RecMind [38] | $66.58_{\pm9.29}$ | $3.77_{\pm0.30}$ | $58.92_{\pm4.72}$ | $4.00_{\pm0.18}$ | $62.82_{\pm5.88}$ | $3.88_{\pm0.19}$ |
| | InteRec [39] | $\underline{70.28}_{\pm10.35}$ | $\underline{3.67}_{\pm0.38}$ | $\underline{61.83}_{\pm6.44}$ | $\underline{3.88}_{\pm0.15}$ | $\underline{66.36}_{\pm5.91}$ | $\underline{3.77}_{\pm0.23}$ |
| | SimTom [37] | $66.20_{\pm9.71}$ | $3.88_{\pm0.30}$ | $52.58_{\pm7.46}$ | $4.17_{\pm0.17}$ | $59.45_{\pm4.37}$ | $4.02_{\pm0.17}$ |
| | AdaPA-Agent | $\mathbf{71.27}_{\pm9.37}$ | $\mathbf{3.64}_{\pm0.30}$ | $\mathbf{63.04}_{\pm4.84}$ | $\mathbf{3.83}_{\pm0.19}$ | $\mathbf{67.24}_{\pm4.41}$ | $\mathbf{3.73}_{\pm0.23}$ |
| 7 | ReAct [16] | $73.50_{\pm7.77}$ | $5.06_{\pm0.39}$ | $51.92_{\pm3.31}$ | $5.59_{\pm0.21}$ | $63.40_{\pm4.60}$ | $5.31_{\pm0.27}$ |
| | Reflexion [17] | $79.70_{\pm5.74}$ | $4.51_{\pm0.43}$ | $62.08_{\pm4.86}$ | $5.20_{\pm0.22}$ | $71.40_{\pm2.73}$ | $4.83_{\pm0.27}$ |
| | RecMind [38] | $\mathbf{81.26}_{\pm5.38}$ | $4.40_{\pm0.43}$ | $65.24_{\pm5.20}$ | $4.82_{\pm0.26}$ | $73.80_{\pm2.93}$ | $4.59_{\pm0.24}$ |
| | InteRec [39] | $79.64_{\pm5.81}$ | $\mathbf{4.26}_{\pm0.47}$ | $\underline{69.60}_{\pm7.84}$ | $\mathbf{4.51}_{\pm0.27}$ | $\underline{75.00}_{\pm3.33}$ | $\mathbf{4.38}_{\pm0.17}$ |
| | SimTom [37] | $79.98_{\pm4.94}$ | $4.44_{\pm0.42}$ | $64.07_{\pm7.01}$ | $5.07_{\pm0.32}$ | $72.60_{\pm3.27}$ | $4.73_{\pm0.30}$ |
| | AdaPA-Agent | $\underline{80.36}_{\pm5.64}$ | $\underline{4.41}_{\pm0.42}$ | $\mathbf{70.45}_{\pm6.21}$ | $\mathbf{4.51}_{\pm0.30}$ | $\mathbf{75.41}_{\pm3.07}$ | $\underline{4.46}_{\pm0.36}$ |

Table 1: Main results of the AdaPA-Agent method and baseline models on the conversational recommendation task. The table presents recommendation successful rate (RSR) and average interaction rounds (AIR) for long-term, short-term, and overall performance, evaluated under different maximum steps (3, 5, and 7).

previous states. **InteRec** [39] integrates long-term and short-term memory mechanisms to enhance personalized recommendations, improving the understanding of user preferences and context for more relevant interactions.

### 5.1.2 Implementation Details

For AdaPA-Agent, we use Llama-3.1-8B-Instruct [40] as the local LLM to generate the next-token distribution. In the interaction-augmentation phase, we set $K = 4$ for the conversational recommendation task and $K = 2$ for the personalized web interaction task. In the preference arithmetic phase, we set $M = 2$ for both two tasks, i.e., we choose top 2 preferences to generate the personalized next-token distribution. In the conversational recommendation task, GPT-4o [41] serves as the user simulator, while DeepSeek V2.5 [42] supports all baselines and AdaPA-Agent. For the personalized web interaction task, GPT-4o is used consistently across all baselines and AdaPA-Agent. To reduce generation randomness, we set the LLM temperature to 0 during all evaluations.

### 5.2 Conversational Recommendation Task

### 5.2.1 Task Introduction

**(1) Task Description**: Conversational recommendation systems personalize suggestions through real-time interactions. Users express needs via natural language, while the agent clarifies and recommends accordingly. To provide recommendations that better align with the user's intent, the agent needs to be able to model the changes in the user's preference strength. **(2) Task Construction**: To evaluate how well our method models preference strength, we divide user movie preferences into long-term and short-term types. Based on the Reddit-Movie dataset [6], we filter 984 unique users with sufficient historical data. Then, we extract their stable preferences from historical data as long-term and select unrelated movies to define short-term preferences. A dynamic simulation environment is built using an LLM-based user simulator [43, 44], where each task randomly assigns a dominant preference type to simulate the conversation. This setup allows controlled variation in preference strength, enabling direct and systematic evaluation of our method. We introduce the setup details of task settings and user simulator in Appendix C. **(3) Evaluation Metrics**: The task is limited to $T$ rounds. If the agent recommends the target movie within $k$ rounds ($k \neq T$), it succeeds; otherwise, it fails. Performance is measured by: *Recommendation Successful Rate (RSR)* =

| Method | Search | | Recommendation | | Review | | Overall | |
|---|---|---|---|---|---|---|---|---|
| | F. Acc | R. Acc | F. Acc | R. Acc | F. Acc | R. Acc | F. Acc | R. Acc |
| Random Memory | 0.974 | 0.640 | 0.296 | 0.018 | 0.996 | 0.442 | 0.745 | 0.357 |
| Last Memory | 0.937 | 0.626 | 0.432 | 0.028 | **1.000** | 0.442 | 0.782 | 0.357 |
| Relevant Memory | 0.928 | 0.622 | 0.492 | **0.030** | **1.000** | 0.443 | 0.800 | 0.356 |
| ReAct [16] | 0.903 | 0.605 | 0.560 | 0.027 | 0.996 | 0.444 | 0.815 | 0.350 |
| RecMind [38] | 0.981 | 0.645 | 0.226 | 0.017 | 0.990 | 0.442 | 0.721 | 0.359 |
| AdaPA-Agent | **0.987** | **0.654** | **0.592** | 0.027 | **1.000** | **0.457** | **0.851** | **0.367** |

Table 2: Main results of the AdaPA-Agent method and baseline models on the single-turn track settings of personalized web interaction. The table shows the function accuracy (F. Acc.) and response accuracy (R. Acc.) for the three web services: Search, Recommendation and Review.

| Method | Search | | | Recommendation | | | Review | | | Overall | | |
|---|---|---|---|---|---|---|---|---|---|---|---|---|
| | F. Acc | R. Acc | Avg. Steps | F. Acc | R. Acc | Avg. Steps | F. Acc | R. Acc | Avg. Steps | F. Acc | R. Acc | Avg. Steps |
| Random Memory | 0.999 | 0.680 | 4.193 | 0.703 | 0.042 | 4.474 | 1.000 | 0.448 | 2.007 | 0.896 | 0.380 | 3.564 |
| Last Memory | 0.996 | 0.676 | 4.229 | 0.708 | **0.045** | 4.252 | **1.000** | 0.449 | 2.007 | 0.897 | 0.381 | 3.597 |
| Relevant Memory | 0.996 | 0.686 | 4.233 | 0.715 | 0.042 | 4.564 | 0.999 | 0.448 | 2.008 | 0.899 | 0.383 | 3.609 |
| ReAct [16] | 0.996 | 0.674 | 4.657 | 0.218 | 0.013 | 5.468 | 0.974 | 0.448 | 2.129 | 0.718 | 0.369 | 4.198 |
| Reflexion [17] | **1.000** | 0.686 | 5.406 | 0.281 | 0.014 | 6.145 | 0.976 | 0.449 | 2.145 | 0.741 | 0.373 | 4.579 |
| RecMind [38] | 0.997 | 0.642 | 6.728 | 0.347 | 0.026 | 6.003 | 0.997 | 0.451 | 2.107 | 0.771 | 0.364 | 4.938 |
| InteRec [39] | 0.999 | 0.642 | 3.110 | 0.618 | 0.022 | 3.008 | 1.000 | 0.447 | 2.001 | 0.867 | 0.362 | 2.706 |
| AdaPA-Agent | 0.999 | **0.698** | 5.352 | **0.768** | 0.039 | 3.485 | **1.000** | **0.455** | 2.004 | **0.917** | **0.386** | 3.592 |

Table 3: Main results of the AdaPA-Agent method and baseline models on the multi-turn track settings of personalized web interaction. The table shows the function accuracy (F. Acc.), response accuracy (R. Acc.) and average steps (Avg. Steps) for the three web services: Search, Recommendation and Review.

$\frac{N_s}{N} \times 100\%$, *Average Interaction Rounds (AIR)* $= \sum_{i=1}^{N} \frac{k_i}{T}$, where $N$ is the total tasks, $N_s$ the successful recommendations, and $k_i$ the interaction rounds in task $i$. Higher RSR and lower AIR indicate better efficiency.

### 5.2.2 Results Analysis

The results in Table 1 compare AdaPA-Agent with baseline models under varying maximum steps (3, 5, and 7) in the conversational recommendation task. AdaPA-Agent consistently outperforms baseline methods in terms of RSR while maintaining competitive step efficiency. AdaPA-Agent maintains a stable and robust performance across both long-term and short-term tasks, unlike some baseline methods that exhibit imbalances. In contrast, baseline methods often show trade-offs. For instance, RecMind performs well on short-term tasks but suffers on long-term preferences, while Reflexion achieves moderate long-term results but lags on short-term tasks. These inconsistencies suggest a lack of adaptive preference modeling. AdaPA-Agent addresses this gap by estimating dynamic preference strengths and adjusting generation accordingly, leading to stable performance across diverse user intents. Although InteRec has explicitly considered the long-term preference and short-term preference, it still suffers the problem of modeling the preference strength and has a lower performance than AdaPA-Agent. Furthermore, AdaPA-Agent strikes an optimal balance between RSR and AIR, making it more efficient in real-time applications. At just 3 steps, it outperforms all baselines on overall RSR (41.45% vs. 38.60% by RecMind and 36.01% by SimTom), and achieves the lowest AIR (2.64). This demonstrates that AdaPA-Agent can effectively estimate preference strengths with minimal feedback and provide accurate recommendations even within short interactions, confirming its effectiveness in real-world conversational recommendation systems.

### 5.3 Personalized Web Interaction Task

### 5.3.1 Task Introduction

**(1) Task Description**: Personalized Web agents infer user preferences to enhance web services. We use the PWABench benchmark [27], which includes tasks like personalized search, recommenda-

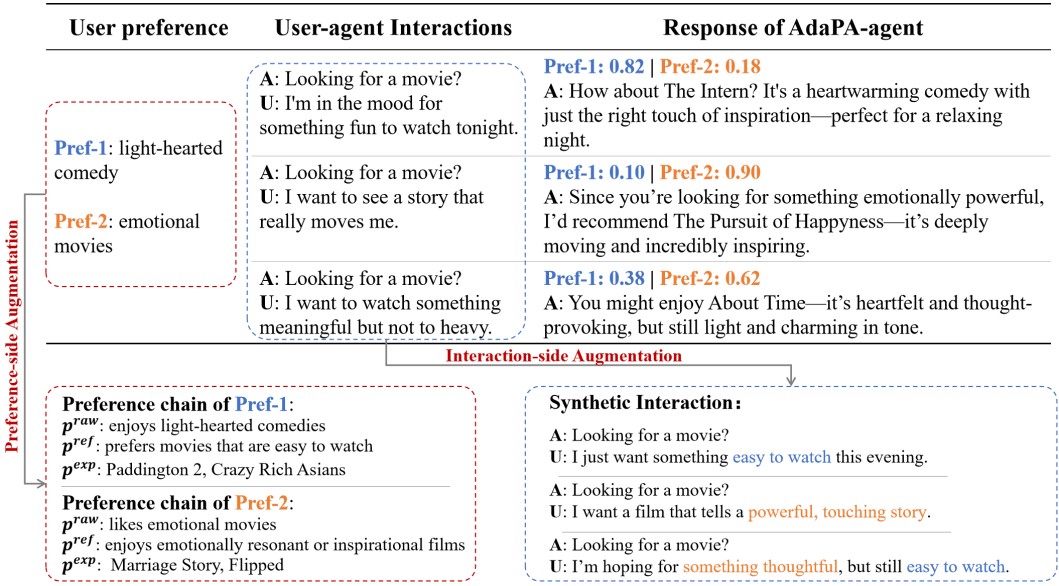

| User preference | User-agent Interactions | Response of AdaPA-agent |
|---|---|---|

**Preference-side Augmentation**

**Pref-1**: light-hearted comedy

**Pref-2**: emotional movies

**A**: Looking for a movie?
**U**: I'm in the mood for something fun to watch tonight.

**Pref-1: 0.82 | Pref-2: 0.18**
**A**: How about The Intern? It's a heartwarming comedy with just the right touch of inspiration—perfect for a relaxing night.

**A**: Looking for a movie?
**U**: I want to see a story that really moves me.

**Pref-1: 0.10 | Pref-2: 0.90**
**A**: Since you're looking for something emotionally powerful, I'd recommend The Pursuit of Happyness—it's deeply moving and incredibly inspiring.

**A**: Looking for a movie?
**U**: I want to watch something meaningful but not to heavy.

**Pref-1: 0.38 | Pref-2: 0.62**
**A**: You might enjoy About Time—it's heartfelt and thought-provoking, but still light and charming in tone.

**Interaction-side Augmentation**

**Preference chain of Pref-1**:
$p^{raw}$: enjoys light-hearted comedies
$p^{ref}$: prefers movies that are easy to watch
$p^{exp}$: Paddington 2, Crazy Rich Asians

**Preference chain of Pref-2**:
$p^{raw}$: likes emotional movies
$p^{ref}$: enjoys emotionally resonant or inspirational films
$p^{exp}$: Marriage Story, Flipped

**Synthetic Interaction：**

**A**: Looking for a movie?
**U**: I just want something easy to watch this evening.

**A**: Looking for a movie?
**U**: I want a film that tells a powerful, touching story.

**A**: Looking for a movie?
**U**: I'm hoping for something thoughtful, but still easy to watch.

Figure 3: Case study of AdaPA-Agent in a movie recommendation scenario. The model dynamically adjusts preference strengths based on user-agent interactions and generates personalized responses.

tion, and review generation. These tasks require LLMs to select the right web function and parameters for personalized outputs. PWABench also provides three memory retrieval baselines–Random Memory, Last Memory, and Relevant Memory–that sample different historical data to assist agents. To better reflect real-world user behaviors, we do not restrict users to only two preferences as in the conversational recommendation task. Instead, each user may have multiple active preferences with diverse and dynamically shifting strengths, posing greater challenges for accurate modeling and adaptation. **(2) Evaluation Metrics**: the evaluation metrics for the system include three key components: (a) *Function Accuracy (F. Acc)*: evaluates the agent's ability to select the correct web function and parameters, scoring 1 for correct selection and 0 otherwise; (b) *Response Accuracy (R. Acc)*: for search and recommendation tasks, this metric uses the rank $r$ of the target product within the returned product list as the performance indicator and is calculated as:

$$\text{R. Acc} = \begin{cases} 1 - \frac{r-1}{10}, & \text{if } r \leq 10 \\ 0, & \text{if } r > 10 \end{cases};$$ (9)

and (c) *Average Steps (Avg. Steps)*: measures total actions needed to complete a task, where fewer steps indicate higher efficiency.

### 5.3.2 Results Analysis

In the personalized web interaction task, accurately understanding user intent requires distinguishing subtle differences in preference-driven behaviore.g., discerning whether a user prefers to search for a product or directly receive a recommendation. This challenge is particularly pronounced in single-turn interactions, where the agent lacks rich dialogue context. As shown in Table 2, AdaPA-Agent achieves the best overall performance across all services, reaching the highest overall function accuracy (F. Acc: 0.851) and response accuracy (R. Acc: 0.367). In particular, AdaPA-Agent significantly outperforms prior methods on recommendation-related interactions (F. Acc: 0.592), a setting where user intent is often implicit and preference strength is most influential. Compared to RecMind (0.226) and Random Memory (0.296), AdaPA-Agent improves recommendation accuracy by over 30 absolute points, showing its ability to infer the dominant preferences even with sparse interaction.

As shown in Table 3, in the multi-turn track, user feedback from multiple rounds is utilized to refine the model's personalized output, leading to performance improvements across all methods compared to the single-turn track. However, AdaPA-Agent stands out not only in achieving higher function accuracy and response accuracy than the baseline methods but also in maintaining relatively small average steps. This highlights the efficient preference modeling capabilities of AdaPA-Agent, as it

does not rely on excessive user feedback. While methods like Reflexion and ReAct, which depend on user feedback to update their understanding of user preferences, require more steps on average.

These results affirm the core advantage of AdaPA-Agent: by modeling preference strengths adaptively, it delivers both more accurate and more efficient personalized servicesespecially in settings where feedback is limited or user intent is ambiguous.

## 5.4 Case Study

We conduct a case study to examine how AdaPA-Agent dynamically adjusts to user preferences in a conversational movie recommendation scenario with two competing preferences: *light-hearted comedy* and *emotional movies*. As shown in Figure 3, across different user utterances, the preference weights inferred by AdaPA-Agent shift in a consistent and interpretable manner, accurately reflecting subtle changes in user intent. For instance, when the user mentions wanting a story that really moves me, the alignment score for the emotional preference sharply increases, prompting a matching recommendation. In contrast, when the user seeks something fun, the model shifts weight toward the light-hearted preference. This illustrates two key strengths of our approach: (1) the ability to capture nuanced preference signals through alignment-based estimation without explicit feedback, and (2) the controllability of generation via preference-weighted decoding, allowing the agent to produce responses that align precisely with the user's evolving priorities. This example highlights how AdaPA-Agent enables fine-grained and interpretable adaptation in real-time personalized interactions.

## 5.5 Additional Experiments

The appendix provides extensive ablation studies (Appendix D) that further validate the effectiveness of AdaPA-Agent in *modeling dynamic preference strengths*. These analyses demonstrate that (1) dual-side augmentation substantially enhances robustness and fine-grained preference estimation (Appendix D.1), (2) the proposed LLM-based alignment scorer more accurately captures semantic alignment between user preferences and interactions compared to embedding-based methods (Appendix D.2), and (3) the continuous preference arithmetic formulation enables more expressive and controllable generation than prompt-based methods (Appendix D.3).

## 6 Conclusion

We present **AdaPA-Agent**, a training-free framework for enhancing LLM-based agents through dynamic preference modeling. AdaPA-Agent addresses two key challenges in personalization: (i) estimating the relative strength of user preferences without relying on explicit user feedback, and (ii) effectively incorporating these strengths into the generation process. To solve the first challenge, we introduce *Alignment-Based Strength Estimation*, which leverages dual-side augmentation and an LLM-based alignment scorer to infer fine-grained preference weights from implicit user behaviors. To solve the second challenge, we propose *Controllable Personalized Generation*, which utilizes the estimated strengths via preference arithmetic to control the personalized generation process. Our experiments on two personalized agent tasks–conversational recommendation and personalized web agent–demonstrate that AdaPA-Agent consistently improves alignment with users' evolving intents and priorities by modeling dynamic preferences.

## Acknowledgement

This work was supported in part by the National Science Fund for Distinguished Young Scholars under Grant 62025602; in part by the National Natural Science Foundation of China under Grant U22B2036, and Grant 11931015; in part by the Technological Innovation Team of Shaanxi Province (No. 2025RS-CXTD-009), the International Cooperation Project of Shaanxi Province (No. 2025GH-YBXM-017); in part by the Fok Ying-Tong Education Foundation, China, under Grant 171105; in part by the Fundamental Research Funds for the Central Universities under Grant D5000230366; in part by the Fundamental Research Funds for the Central Universities under Grant G2024WD0151, and in part by the Tencent Foundation and Xplorer Prize. Y. Wang is sponsored by Beijing Nova Program. Q. Yao is sponsored by CCF-Zhipu Large Model Innovation Fund (No. Zhipu202402).

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

# A Prompts for Alignment-Based Strength Estimation

The prompt 1 is used for *preference-side augmentation*, a key step in estimating dynamic preference strengths. It constructs structured preference chains that represent user preferences across three semantic levels: from abstract to concrete. The model is instructed to think step by step and output the structured result strictly in JSON format.

---

**Prompt 1: Reasoning Augmentation**

Your goal is to help an AI agent better understand user preferences for personalized response generation.
Given the following user preference description:
{USER_PREFERENCE_CONTEXT}

Your task is to construct a structured preference chain with three levels of semantic granularity.

Please think step by step and provide:
1. raw: the original or high-level form of the preference
2. refined: a context-aware, clearer reformulation of the raw preference
3. example: representative items, actions, or behaviors that reflect the preference

Output the result in the following JSON format:
{
"raw": "<fill in raw preference>",
"refined": "<fill in refined preference>",
"example": ["<item_1>", "<item_2>", "..."]
}

---

This prompt is used for *interaction-side augmentation*, a key step in estimating dynamic preference strengths. It instructs the LLM to generate $K$ paraphrased versions of a given user-agent interaction $\mathcal{D}_t^u$, maintaining the same structure and intent but varying the surface-level expressions. This helps expand the range of observable user behaviors, reduces scoring bias from any single expression, and enables more robust alignment between user intent and preferences during strength estimation.

---

**Prompt 2: Intuiton Augmentation**

Please generate {K} semantically equivalent but lexically diverse conversations based on the following user-agent interaction:
User-Agent Interaction:
{$\mathcal{D}_t^u$}

Instructions:
1. Return only the {K} generated interactions, each preserving the number of turns and speaker roles.
2. Keep the user's underlying intent consistent with the original conversation.
3. Use varied wording and sentence structures to enhance linguistic diversity.

---

This prompt is designed for use in the *alignment scoring stage* of our framework, where an LLM is asked to assess how well a given user-agent interaction reflects a specific user preference. The input includes a structured preference chain (comprising raw, refined, and example-level descriptions) and the current interaction between user and agent. The LLM is instructed to reason step-by-step, interpret the semantics of the preference, and judge the degree of alignment exhibited in the dialogue. It outputs a single numerical alignment score ranging from 0 to 10, where higher values indicate stronger semantic alignment. This score serves as the foundation for computing the relative strength of each preference in the final generation stage.

**Algorithm 1** Adaptive Preference Arithmetic (AdaPA-Agent)

---

**Require:** Accumulated user interaction data $\mathcal{D}_t^u$ up to time $t$,
    User preference set $\mathcal{P}^u = \{p_i^u\}_{i=1}^M$,
    Chain-of-thought generator $\text{CoT}(\cdot)$,
    Interaction paraphraser $\mathbf{g}(\cdot)$,
    Alignment scorer $\mathbf{f}(\cdot, \cdot)$
**Ensure:** Personalized agent response $\mathcal{O}_t^u$

1: **## 1: Alignment-Based Strength Estimation**
2: $\mathcal{A}^u \leftarrow \mathbf{g}(\mathcal{D}_t^u) \cup \{\mathcal{D}_t^u\}$          # Interaction-side augmentation
3: **for** $i = 1$ **to** $M$ **do**
4:     $\mathcal{C}_i \leftarrow \text{CoT}(p_i^u)$          # Preference-side augmentation
5:     $s_i \leftarrow \frac{1}{|\mathcal{A}^u|} \sum_{\mathcal{D} \in \mathcal{A}^u} \mathbf{f}(\mathcal{C}_i, \mathcal{D})$          # Alignment scoring
6: **end for**
7: Normalize: $\omega_i \leftarrow \frac{s_i}{\sum_{j=1}^M s_j}$ for $i = 1{:}M$          # Strength weighting

8: **## 2: Controllable Personalized Generation**
9: Initialize: $\mathcal{O}_t^u \leftarrow [\,]$          # Empty response buffer
10: **while** $w \neq \texttt{EOS}$ **do**
11:     $\mathbf{P}_{\text{mix}} \leftarrow \sum_{i=1}^M \omega_i \cdot \text{LLM}\big((\mathcal{O}_t^u, \mathcal{D}_t^u, \mathcal{C}_i)\big)$
12:     $w \sim \mathbf{P}_{\text{mix}}$          ## Sample next token
13:     $\mathcal{O}_t^u \leftarrow [\mathcal{O}_t^u, w]$          ## Update response prefix
14: **end while**

15: **return** $\mathcal{O}_t^u$

---

> **Prompt 3: Measuring Correlation Score**
>
> You are an alignment evaluator tasked with assessing how well a user-agent interaction aligns with a given preference chain.
>
> Given:
> - User-Agent Interaction: $\{\mathcal{D}_t^u\}$
> - Preference Chain: $\{\mathcal{C}_i\}$, which includes a raw description, a refined version, and representative examples of a specific user preference.
>
> Instructions:
> 1. Think step by step.
> 2. Judge how strongly the interaction reflects the intent or semantics of the given preference.
> 3. Provide a single alignment score between 0 and 10:
> - 10 = perfectly aligned (preference is clearly implied or reflected)
> - 0 = completely unrelated
> 4. Only return the numerical score.

## B   Algorithm for AdaPA-Agent

Algorithm 1 presents the core procedure of **Adaptive Preference Arithmetic (AdaPA-Agent)**, which progressively adjusts the influence of user preferences based on the accumulated interaction context to generate personalized responses. The algorithm operates in two stages that jointly enable adaptive, context-aware preference control.

- **1. Alignment-Based Strength Estimation (Lines 1-7):** At each interaction step $t$, the agent updates its internal record of user interactions $\mathcal{D}_t^u$. To enrich contextual diversity, it performs *interaction-side augmentation* by paraphrasing the interaction history via $\mathbf{g}(\cdot)$, forming an expanded set $\mathcal{A}^u$ (Line 2). For each user preference $p_i^u \in \mathcal{P}^u$, the agent generates reasoning traces using a CoT generator (Line 4), then evaluates the semantic alignment between each preference trace and the augmented interaction set using the alignment scorer $\mathbf{f}(\cdot, \cdot)$ (Line 5). The resulting alignment scores $\{s_i\}$ are normalized into preference

weights $\{\omega_i\}$ (Line 7), representing each preferences contextual relevance within the ongoing dialogue. This ensures that the model adaptively reflects preference shifts as user intent evolves over multiple turns.

- **2. Controllable Personalized Generation (Lines 8-15):** Initialized with an empty output prefix (Line 9), the agent generates a response token by token. At each step, it combines next-token distributions weighted by the computed $\omega_i$ across all preferences (Line 11), samples the next token (Line 12), and appends it to the output (Line 13), repeating until the end-of-sequence (EOS) token is reached.

By continually re-estimating preference strengths and integrating them into generation, AdaPA-Agent achieves *context-aware, progressive adaptation* across interaction turns. This allows the agent to maintain alignment with evolving user intent without requiring retraining or explicit feedback, resulting in more personalized and consistent dialogue behavior.

---

**Prompt 4: LLM-Based User Simulator for Conversational Recommendation**

You will play the role of a user interacting with a conversational movie recommendation system. Your task is to find a movie that matches your current taste, which is influenced by your preferences.

**Role & Behavior Guidelines:**

- Engage naturally with the agent by gradually revealing your preferences.
- Focus only on requesting or evaluating movie suggestions based on your preferences.
- **Never** mention the name of your target movie.

**Task Information:**

- In this task, your preference is: {prefer_info}.
- In this task, your target movie is: {target_item}.

**Simulation Rules:**

1. Start with vague intent (e.g., "I want to watch something meaningful").
2. Reveal preference cues as the agent asks follow-up questions.
3. Accept recommendations if they match the target movie.
4. Politely reject unrelated ones and give vague but helpful feedback (e.g., "That's not quite what I'm looking for").
5. Maintain a natural and preference-driven tone throughout.

**IMPORTANT**: Your role is to simulate a movie enthusiast who is exploring potential movie recommendations, not to reveal the exact title of the target movie–{target_item}. Keep the conversation natural and engaging, and always focus on requesting recommendations or giving feedback based on the suggestions you receive.

---

## C   Details of Conversation Recommendation Task Construction

To carefully evaluate our method's ability to model preference strength, we categorize user movie preferences into two types: long-term and short-term. This distinction allows for controlled evaluation of how well the method adapts to shifts in preference strength. The dataset of the task is based on Reddit-Movie [6], and we extract stable preferences from user's historical data as long-term preferences and generate a corresponding movie list. We select several movies unrelated to long-term preferences and regard their features as the user's short-term preferences.

Furthermore, we create a dynamic simulation environment where the agent interacts with a LLM-based user simulator. For each recommendation task, the user simulator randomly selects which type of preference (long-term or short-term) will dominate the interaction, allowing us to control the variation of preference strength. This enables us to directly manipulate and evaluate the model's response to dynamic shifts in user preferences, providing a clear validation framework for our method. To ensure the reliability of this simulation setup, our user simulator design follows common practices in recent literature [43, 44, 45, 46]. Specifically, we inject user attributes and interaction rules into prompt templates, and the LLM dynamically generates responses based on the accumulated dialogue context, enabling evolving user behavior over turns. We construct the user simulator for

the conversation recommendation task through the following prompt template (Prompt 4). To avoid exposing the target movie, we insert the rule that the LLM should not mention the target movie in the conversation. Besides, we use regular expressions in the code to detect and mask any accidental exposure (replacing it with "*") during the interaction between the LLM agent and the user simulator. Although our simulator may not perfectly replicate real users, it provides a controlled, consistent, and adaptive framework for evaluating the effectiveness of preference modeling strategies. To reduce generation randomness, we set the LLM temperature to 0 during all evaluations. For each reported result in Table 1, we ran 10 trials with different random seeds and computed the standard deviation across these runs.

# D    Ablation Studies of AdaPA-Agent

## D.1    Effectiveness of Dual-side Augmentation

To assess the contribution of dual-side augmentation, we compare three variants of AdaPA-Agent in preference strength estimation: using both preference- and interaction-side augmentation (w/ D-side), using only preference-side augmentation (w/ P-side), and without any augmentation (w/o D-side). To validate dual-side augmentation, we simplified the conversational recommendation and personalized web interaction tasks. In the former, we use binary classification to predict whether a user's desired movie aligns with long-term or short-term preferences. In the latter, we use three-class classification to determine if the required service relates to search, recommendation, or review. As shown in Figure 4, the dual-side setup achieves the highest win rate in both tasks—77.3% in conversational recommendation and 63.8% in personalized web interaction—significantly outperforming the other variants. These results highlight that both augmentation dimensions are critical: while preference-side augmentation improves semantic richness, interaction-side augmentation enhances robustness against linguistic variability.

Relying solely on one side introduces bias or sparsity, whereas combining both provides more comprehensive signals for modeling fine-grained preference strengths. By contrast, dual-side augmentation enables AdaPA-Agent to capture both semantic intent and linguistic variability, significantly improving the accuracy of preference strength estimation.

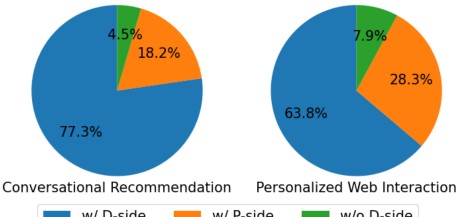

Conversational Recommendation       Personalized Web Interaction

■ w/ D-side    ■ w/ P-side    ■ w/o D-side

Figure 4: Win rates of AdaPA-Agent on using dual-side data augmentation (w/ D-side), only preference-side augmentation (w/ P-side) and without using any data augmentation (w/o D-side) in preference strengths estimation.

| $K$ | Recall | Precision | F1-Score |
|---|---|---|---|
| 0 | 25.32 | 70.21 | 37.22 |
| 1 | 43.73 | 66.70 | 52.83 |
| 2 | 56.27 | 72.43 | 63.34 |
| 3 | 68.79 | 75.86 | 72.15 |
| 4 | **87.50** | **90.32** | **88.89** |
| 5 | 78.13 | 80.64 | 79.35 |
| 6 | 71.88 | 71.88 | 71.88 |

Table 4: Effect of varying the number of generated user-agent interactions ($K$) on the binary classification results of the conversational recommendation task.

Building upon the experimental setup in Figure 4, we further investigate the effect of interaction-side augmentation on dynamic preferences modeling. As shown in the Table 4, $K$ represents the amount of data augmentation, while recall, precision, and F1-score indicate the model's performance on binary classification for different levels of data augmentation. We observe that as $K$ increases, the recall significantly improves, showing that more augmentation helps identify relevant samples. Similarly, precision and F1-score also increase, reaching their peak values at $K = 4$, after which they start to decrease as $K$ continues to rise. These results show that, the initial improvement stems from the augmented interaction data providing useful preference information, but excessive augmentation introduces noise which results in poor performance. The same phenomenon also occurs in the personalized web interaction task.

## D.2    Analysis of LLM-based Alignment Scorer

To evaluate the effectiveness of estimating preference-interaction alignment scores, we compare our proposed LLM-based method with a traditional embedding-based baseline using Sentence-BERT.

| Maximum Steps | Embedding-based | LLM-based (ours) |
|:---:|:---:|:---:|
| 3 | 35.39 | 41.45 |
| 5 | 58.56 | 70.45 |
| 7 | 67.58 | 75.41 |

Table 5: Recommendation success rate (RSR) of conversational recommendation using either an embedding-based alignment scorer (Sentence-BERT) or our proposed LLM-based alignment scorer. The LLM-based method consistently outperforms the embedding baseline across different maximum step limits, indicating its superior ability to estimate alignment between user preferences and user-agent interactions.

For the baseline, we encode both the preference description and the user-agent interaction with Sentence-BERT and treat the cosine similarity of the two embeddings as the alignment score. As shown in Table 5, the LLM-based alignment scorer consistently outperforms the embedding-based method across varying maximum step limits in the conversational recommendation task. This performance gap suggests that LLMs are better at capturing the nuanced alignment between user preferences and interaction trajectories. Unlike static embedding models, LLMs can perform contextual reasoning and infer implicit preference signals, leading to more reliable estimation of preference strength and, ultimately, more accurate and intention-aligned recommendations.

---

**Prompt 5: Prompt Engineering**

Here are the user's long-term and short-term preferences and their weights (0 to 1, where 0 means no relevance and 1 means the highest relevance):
User's long-term preference: {}
Long-term preference weight: {}
User's short-term preference: {}
Short-term preference weight: {}
Please respond to the user's query based on the long-term and short-term preferences and their weights.
User query: {}

---

### D.3 Effectiveness of Preference Arithmetic

Figure 5 demonstrates the effectiveness of the preference arithmetic method in generating personalized responses based on varying strength combinations $(\omega_S, \omega_L)$ of short-term and long-term preferences , compared to the traditional prompting engineering approach. Both preference arithmetic and prompt engineering consider two type of preferences and their weights, while the former uses weights as coefficients in arithmetic, the latter regards them as prompts. We use Llama-2-7b-chat as the backbone for these two method, while use GPT-4o to generate the ground truth. The semantic similarity between the generated content and the ground truth is calculated by Sentence-BERT. From the figure, it is evident that preference arithmetic consistently outperforms prompting engineering in terms of alignment

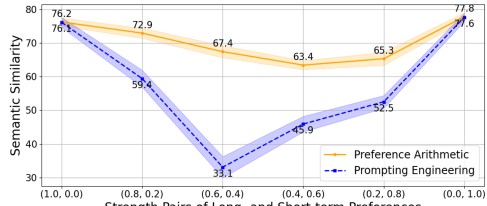

Figure 5: Performance comparison of the preference arithmetic and prompting engineering methods in generating personalized responses based on varying pairs of $(\omega_S, \omega_L)$. The figure shows the alignment between generated content and ground truth generated by GPT-4o.

with the ground truth. This indicates that adjusting preference-conditioned next-token distributions using preference strengths more effectively controls the influence of long- and short-term preferences on the generated content. Moreover, the preference arithmetic method is more accurate in distinguishing between strength differences, particularly when the strengths of long- and short-term preferences are close, such as in the (0.4, 0.6) strength pair. While the traditional prompting engineering method struggles with this, likely due to the limitations of understanding of numerical information in text. This ability allows preference arithmetic to generate more consistent personal-

ized responses compared to prompting engineering. The prompt engineering template is designed as follows (Prompt 5):

We further compare AdaPA-Agent with a baseline method, termed **Semantic Strength Prompting (SSP)**, where the strengths of preferences are represented as five discrete textual descriptors (*weakest*, *weak*, *neutral*, *strong*, *strongest*). We evaluate both methods on the **Conversational Recommendation Task**, using Recommendation Success Rate (RSR) and Average Interaction Rounds (AIR) as metrics. As shown in Table D.3, AdaPA-Agent consistently outperforms the SSP baseline across all interaction settings. Although such discrete verbal schemes may appear expressive, they are inherently limited to a finite number of strength categories, leading to coarse control granularity. In contrast, AdaPA-Agent models preference strengths as continuous values within $[0, 1]$, enabling infinitely many combinations of strength settings. This formulation allows more nuanced and personalized control over how preferences influence generation. Moreover, since verbal descriptors like *weak* or *strong* can be semantically ambiguous and inconsistently interpreted by LLMs, token-wise distribution mixing provides explicit and interpretable control by directly manipulating strengths over next-token distributions, thereby reducing ambiguity and semantic drift. Therefore, these results demonstrate that AdaPA-Agent achieves superior expressivity and controllability compared to discrete strength prompting, enabling more robust and interpretable preference-conditioned generation.

| Max Steps | Method | RSR | AIR |
|---|---|---|---|
| 3 | **AdaPA-Agent** | **41.45** | **2.64** |
|   | SSP | 34.20 | 2.80 |
| 5 | **AdaPA-Agent** | **67.24** | **3.73** |
|   | SSP | 59.60 | 4.13 |
| 7 | **AdaPA-Agent** | **75.41** | **4.46** |
|   | SSP | 70.82 | 4.79 |

Table 6: Performance comparison between AdaPA-Agent and Semantic Strength Prompting (SSP) on the conversational recommendation task.

# E   Experiments Compute Resources

In this work, all experiments are conducted on a machine with NVIDIA A6000*2 GPUs, each GPU has 48G memory.

# F   Limitations

While our framework shows promising results, it has several limitations that open directions for future work. First, our evaluation relies on simulated user interactions, which may not fully capture the nuance of real-world behaviorfuture work could include human-in-the-loop or live deployment studies to validate robustness. Second, the use of multiple LLM calls for alignment scoring increases computational costs, suggesting the need for more efficient approximation techniques. However, given the ongoing decrease in API call costs, we believe the trade-off between API usage and performance improvement in our method is worthwhile. Third, we focus on two domainsmovie recommendation and web interactionwhile extending the approach to broader tasks (e.g., education, health advice) remains unexplored.

# G   Broader Impacts

This work explores dynamic preference modeling for personalized LLM agents, which can enhance user experience in recommendation, web assistance, and interactive applications. By adaptively aligning with user intent without requiring explicit feedback, the proposed method supports more natural and efficient human-AI interactions. However, personalization systems must be carefully designed to avoid reinforcing user biases, exposing sensitive preferences, or leading to over-reliance on AI decisions. Future work should consider fairness, privacy, and user control in the deployment of adaptive preference-aware agents.

