# OpenReview forum: "Adaptive Preference Arithmetic: A Personalized Agent with Adaptive Preference Arithmetic for Dynamic Preference Modeling"
_NeurIPS.cc/2025/Conference — NeurIPS 2025 poster_

### Official Review · Reviewer_vYZm · 2025-06-09

**Clarity:** 3
**Significance:** 3
**Originality:** 3
**Rating:** 4
**Confidence:** 4

**Summary:**

This paper proposes AdaPA-Agent, a framework for personalizing large language model (LLM) agents by dynamically estimating and leveraging the relative strengths of multiple user preferences. Rather than relying on explicit user feedback, AdaPA-Agent infers preference strengths from user-agent interaction history using a combination of dual-side augmentation (enriching both preference representations and interaction paraphrases) and an LLM-based alignment scoring mechanism. The estimated strengths are then used to guide the agent’s responses by linearly combining the next-token distributions conditioned on each preference. Experiments on conversational recommendation and personalized web interaction benchmarks show that AdaPA-Agent achieves higher personalization quality and efficiency compared to several strong baselines, without requiring additional model retraining or explicit user input.

**Questions:**

Please see weaknesses.

**Ethical Concerns:**

["NO or VERY MINOR ethics concerns only"]

**Final Justification:**

The authors' responses have successfully resolved most of my initial concerns. However, I am maintaining my current score because the manuscript, in its present form, is not sufficiently self-contained. I believe a major revision is necessary to integrate crucial details into the main paper, thereby improving its completeness and reproducibility. I am willing to reconsider my evaluation upon seeing a revised version that incorporates these changes.

**Limitations:**

Yes.

**Paper Formatting Concerns:**

The appendix appears before the references section in this submission. Please revise the order to place the references before the appendix, as specified in the [NeurIPS 2025 LaTeX style file](https://media.neurips.cc/Conferences/NeurIPS2025/Styles.zip).

**Quality:**

3

**Strengths And Weaknesses:**

**Strengths**

**S1.** The paper proposes a training-free framework that models dynamic user preference strengths. This approach is highly practical and resource-efficient as it does not require costly fine-tuning or model retraining, making it a flexible solution for adapting to user preferences in practice.

**S1.** The paper proposes a novel approach to estimating preference strengths by combining dual-side augmentation (preference chain construction and interaction paraphrasing) with LLM-based alignment scoring. This enables the extraction of implicit preference signals from dialogue without requiring explicit user feedback.

**S2.** The method demonstrates substantial and consistent performance gains over competitive baselines such as ReAct, Reflexion, and RecMind on both conversational recommendation (Reddit-Movie) and personalized web agent (PersonalWAB) benchmarks. Improvements are shown in both personalization quality (e.g., recommendation success rate) and efficiency (e.g., fewer interaction rounds).

**Weaknesses**

**W1. Adaptivity to dynamic preferences.** While Figure 3 and Section 5.4 show changes in inferred preference strengths, this does not demonstrate true adaptivity to dynamic preference shifts. In my understanding, the framework simply re-computes strengths from scratch at each interaction—functionally equivalent to applying a static estimator repeatedly—rather than maintaining or updating preferences in an adaptive way. As a result, the method passively reflects recent inputs instead of adaptively detecting or responding to preference shifts within a session. Could the authors clarify how AdaPA-Agent can *detect* or quickly adapt to user preference shifts within an *ongoing* multi-turn conversation, rather than simply re-estimating preferences after each isolated interaction?

**W2. Omission of PUMA baseline.** While the paper uses the PersonalWAB [(Cai et al., 2024)](https://arxiv.org/abs/2410.17236) for evaluating personalized web agent performance, it does not include PUMA—the core approach introduced with PersonalWAB—in its baselines. This omission is not discussed or justified. Since PUMA is specifically designed for this benchmark and serves as a state-of-the-art method, a fair comparison requires evaluating AdaPA-Agent against PUMA on the same tasks. The absence of this baseline makes it difficult to assess true progress over prior work; the paper should either include direct comparisons to PUMA or clearly justify its exclusion.

**Minor comments**

**C1. Additional related works.** The controllable personalized generation proposed here resembles recent methods such as [Zhu et al. (2025)](https://arxiv.org/abs/2502.14204) and [He et al. (2024)](https://arxiv.org/abs/2405.01768), but the paper does not discuss similarities or differences. Clearer positioning relative to these works would strengthen the contribution. Also, recent studies on LLM personalization via preference strength estimation, such as [Oh et al. (2024)](https://arxiv.org/abs/2411.00524) and [Hwang et al. (2024)](https://arxiv.org/abs/2312.09337), could be included in Section 2.2 to provide more accurate context. Lastly, “LLM Personalization” seems a quite broad category; I recommend narrowing the scope for a more explicit and meaningful comparison.

**C2. Robustness of CoT-based augmentation.** The CoT-based augmentation process relies on LLM-generated examples to concretize user preferences. However, these examples may not always align with the user’s true preferences (e.g., “prefer healthy food” may not always mean “prefer salads”), introducing spurious associations. Additional clarification, analysis, or robustness experiments regarding this aspect would strengthen the reliability and stability of the proposed framework.

**C3. Ambiguity in preference dimensions.** While the manuscript states that the authors considered "multiple active preferences" for web tasks, it does not specify the exact number or types of these preferences. For the paper to be self-contained, it is beneficial to clearly describe the preference dimensions used in the experiments.

**References**

- Zhu et al. (2025). *On-the-fly Preference Alignment via Principle-Guided Decoding*
- He et al. (2024). *Context Steering: Controllable Personalization at Inference Time*
- Oh et al. (2024). *Comparison-based Active Preference Learning for Multi-dimensional Personalization*
- Hwang et al. (2024). *Promptable Behaviors: Personalizing Multi-objective Rewards from Human Preferences*
- Cai et al. (2024). *Large Language Models Empowered Personalized Web Agents*

---

> ### Author Rebuttal · Authors · 2025-07-31
>
> > **W1.** **Adaptivity to dynamic preferences.** The method passively reflects recent inputs instead of adaptively detecting or responding to preference shifts within a session. Could the authors clarify how AdaPA-Agent can *detect* or quickly adapt to user preference shifts within an *ongoing* multi-turn conversation, rather than simply re-estimating preferences after each isolated interaction?
>
> **A1.** Thank you for the insightful comment. We would like to clarify a possible misunderstanding about the adaptive mechanism in our framework.
>
> While our Alignment-Based Strength Estimation module does re-compute preference strengths at each interaction step, this **re-estimation is based on the full accumulated user-agent interaction records within the session.**
> In other words, the preference strength estimation is continuously updated in response to the **evolving multi-turn conversation**, not based solely on isolated inputs. This design allows the model to **adaptively reflect preference shifts** as they emerge across turns, rather than relying on static or fixed preference encodings.
>
> We briefly outline the adaptive process below for clarity:
>
> 1. For the task at time $t$, as the number of interaction rounds increases, new interaction data will be accumulated in $D_t^u$.
> 2. The updated $D_t^u$ is used to re-compute the strength of each preference $p_i^u \in P^u$ through dual-side augmentation and LLM-based scoring (see Section 4.1).
> 3. The re-computed strengths $\omega_i^{(t)}$ directly guide the generation of the next agent response.
>
> This mechanism ensures that the model does not merely reflect recent utterances, but rather performs **context-aware, progressive adaptation** throughout the session. We will revise Algorithm 1 to make this aspect of the framework clearer.
>
> > **W2.** **Omission of PUMA baseline.** While the paper uses the PersonalWAB for evaluating personalized web agent performance, it does not include PUMA—the core approach introduced with PersonalWAB—in its baselines. This omission is not discussed or justified.
>
> **A2.** Thank you for pointing this out. We acknowledge the importance of comparing against strong baselines, and we would like to clarify why **PUMA was not included in our evaluation**.
>
> Our goal in this paper is to evaluate and compare **agent methods for preference modeling that are training-free and generalizable across multiple personalized agent tasks**. Toward this goal, we deliberately selected **baseline methods that do not rely on fine-tuning or benchmark-specific supervision**, ensuring fair and consistent evaluation in both conversational recommendation and personalized web interaction settings.
>
> The exclusion of PUMA is motivated by the following two reasons:
>
> 1. **Fairness in evaluation:**
>    PUMA is a **fine-tuned model tailored specifically for the PersonalWAB benchmark**, and both PUMA (GPT-4o) and PUMA (LLaMA-7B) variants involve supervised fine-tuning. For instance, they fine-tune LLaMA-2-7B to identify the correct web function, and PUMA (LLaMA-7B) further applies DPO to optimize parameter generation of functions. In contrast, **our method—as well as all selected baselines like ReAct, Reflexion, RecMind, etc.—are training-free** and do not rely on any task-specific gradient updates. Comparing training-free methods directly with benchmark-specific fine-tuned models would not be methodologically fair.
>
> 2. **Generality across domains:**
>    PUMA is **tightly coupled to the PersonalWAB benchmark** and is not designed to generalize to other personalized agent tasks. In contrast, our proposed AdaPA-Agent framework is task-agnostic and has been evaluated across **multiple scenarios**, including both personalized web interaction and conversational recommendation, without task-specific adaptation. As such, our focus is on **cross-task generality and dynamic preference modeling**, which PUMA does not support.
>
> In summary, we deliberately focus on **training-free, plug-and-play personalization frameworks** that are more practical in real-world multi-task deployments. We will add a clarification in the paper to explicitly justify the omission of PUMA in the baseline comparison.
>
> > **C1.** **Additional related works.** The paper lacks discussion of related works on controllable generation (e.g., Zhu et al., He et al.) and preference strength modeling (e.g., Oh et al., Hwang et al.). A clearer positioning and narrower scope within “LLM Personalization” would improve the contribution.
>
> **A3.** We thank the reviewer for the valuable suggestions. We briefly summarize the relevant works and clarify how our approach differs:
>
> * **Zhu et al. (2025)** and **He et al. (2024)** propose controllable text generation methods for LLM that rely on **explicit preference instructions** to guide generation.
> * **Oh et al. (2024)** infers preferences through **multiple rounds of explicit user feedback**, while **Hwang et al. (2024)** models preference-conditioned reward adjustment for **robot policy control**.
>
> In contrast, our method targets **LLM-agent personalization**, with the goal of **dynamically modeling and leveraging preference strength** based solely on observed user-agent interactions—**without requiring additional user feedback/instruction**.
>
> We agree that these works are technically relevant, and we will revise **Section 2.2** to include a **focused discussion** of their similarities and differences. We also appreciate the reviewer’s suggestion to narrow the scope of the "LLM Personalization" category and will adjust the structure accordingly to improve clarity and positioning.
>
> > **C2.** **Robustness of CoT-based augmentation.** The CoT-based augmentation process relies on LLM-generated examples to concretize user preferences. However, these examples may not always align with the user’s true preferences (e.g., “prefer healthy food” may not always mean “prefer salads”), introducing spurious associations. Additional clarification, analysis, or robustness experiments regarding this aspect would strengthen the reliability and stability of the proposed framework.
>
> **A4.** Thank you for this insightful observation. We agree that relying on LLMs to expand preferences from abstract to concrete expressions introduces the risk of unintended or overly narrow interpretations. However, our **design of CoT-based preference chains is intended to mitigate this** by:
>
> 1. **Expanding preferences at multiple semantic levels** (raw → refined → examples) rather than relying on a single point interpretation (Appendix A, Prompt 1). This allows the model to **capture a broader and more nuanced representation** of user intent, rather than reducing it to a single canonical form like “salads.”
>
> 2. We **generate a diverse set of representative examples** (i.e., multiple concrete items per preference) to avoid overfitting to any one interpretation. For example, as shown in Figure 2, a preference like “healthy food” generates a list such as [grilled chicken, salads, avoids greasy], improving robustness. More examples are illustrated in Figure 3.
>
> 3. In **Appendix D.1**, we show through ablation experiments that Preference-side augmentation ("P-side" in Figure 4) **significantly improves accuracy of preference type prediction** across both tasks. This empirically supports the effectiveness of using the CoT-based preference chain.
>
> > **C3.** **Ambiguity in preference dimensions.** While the manuscript states that the authors considered "multiple active preferences" for web tasks, it does not specify the exact number or types of these preferences. For the paper to be self-contained, it is beneficial to clearly describe the preference dimensions used in the experiments.
>
> **A5.** Thank you for pointing this out. We agree that clearly specifying the preference setup can improve clarity and self-containment.
>
> As stated in **Section 5.1.2 Implementation Details (lines 235–237)**, for both the **conversational recommendation** and **personalized web interaction** tasks, **we construct two preferences per user (i.e., $M = 2$)** to serve as the input preference set $P^u$.
>
> * In the **conversational recommendation** task, we **manually simulate** one **long-term preference** (extracted from historical movie interaction data) and one **short-term preference** (constructed by injecting a transient interest). This setup is designed to evaluate the model’s ability to adapt to shifts in dominant preference strength.
>
> * In the **personalized web interaction** task, we **do not predefine specific dimensions** (e.g., long-term or
> short-term). Instead, for each user $u$, we use an LLM to **summarize only two representative shopping preferences** from their real-world historical behavior (e.g., purchase or review records) in the PUMAbench dataset. This design helps to reduce potential noise and ensure the extracted preferences reflect shopping interests depending on the user's history.
>
> We will revise the manuscript to make these details more explicit in the main text, ensuring that the paper remains self-contained.

---

> > ### Comment · Reviewer_vYZm · 2025-08-05
> >
> > Thank you to the authors for the thoughtful and detailed rebuttal.
> >
> > **A1 and A3-5.** I appreciate the authors’ efforts to address these points, and I find the planned updates (e.g., clarifying the adaptive mechanism, expanding related work, highlighting ablation results, and specifying preference setups) to be appropriate and helpful.
> >
> > **A2.** I agree that using PUMA as a direct baseline may be inappropriate, given its benchmark-specific fine-tuning. However, I view PUMA as a valuable reference point that represents an approximate upper bound of performance on PersonalWAB. Including a relative comparison—without necessarily treating it as a baseline—could be beneficial in assessing the practical effectiveness of the proposed method. It would help readers better understand where the proposed training-free framework stands in relation to strong task-specific systems.

---

> > > ### Author Response · Authors · 2025-08-05
> > > **Response to A2 Follow-up**
> > >
> > > We sincerely thank the reviewer for the constructive follow-up. We fully agree that while PUMA is not directly comparable due to fine-tuning on benchmark-specific data, it serves as a valuable reference point, especially as an approximate upper-bound for task-specific performance on PersonalWAB.
> > >
> > > In our revision, we will explicitly include a discussion of PUMA's performance as an upper bound in the results section. This will help readers better assess the practical effectiveness of our training-free method relative to benchmark-optimized systems.
> > >
> > > We appreciate the reviewer’s comments. Your insightful feedback has greatly helped us improve the completeness and clarity of our paper. Thank you again!

---

### Official Review · Reviewer_E4dG · 2025-06-13

**Clarity:** 2
**Significance:** 2
**Originality:** 2
**Rating:** 2
**Confidence:** 3

**Summary:**

This paper introduces a novel framework for large language model (LLM)-based agent personalization called AdaPA-Agent, which achieves more fine-grained personalized response generation by modeling the dynamic strengths of user preferences. The authors point out that although existing methods can already capture the content of user preferences, they fail to effectively model how the relative strengths of these preferences evolve over time. To address this, they propose "Adaptive Preference Arithmetic," which consists of two key modules. The first module, Alignment-Based Strength Estimation, automatically estimates preference strengths from existing user-agent interactions without requiring additional user feedback, using dual-side augmentation and an LLM-based alignment scorer. The second module, Controllable Personalized Generation, controls the generation process by linearly combining next-token distributions conditioned on different preferences, weighted according to their estimated strengths. The framework is evaluated on two tasks—conversational recommendation and personalized web interaction—and the results demonstrate that AdaPA-Agent outperforms baseline methods such as ReAct across multiple metrics, achieving better alignment with user needs when intentions change over time.

**Questions:**

Why is this weighting approach in Eq. 5 effective? Is it possible to find a theoretical basis behind it?

**Ethical Concerns:**

["NO or VERY MINOR ethics concerns only"]

**Limitations:**

Although the "Broader Impacts" section is mentioned in the appendix, it is relatively brief and lacks in-depth analysis of potential risks such as privacy leakage, preference amplification bias, and algorithm misuse.

**Quality:**

2

**Strengths And Weaknesses:**

Stregthens
1. The authors propose a training-free approach to align with dynamic human preferences for personalization, enabling better alignment with user needs when intentions change.
2. AdaPA-Agent is a training-free method that is applicable to various LLM architectures, offering strong versatility and deployment flexibility.

Weakness
1. The method is overly intuitive and experience-based, lacking theoretical foundations.
2. The method calculates preference scores through model computations, but the reliability of these scores from large language models has not been sufficiently validated.
3. Using the CoT prompting approach, the reasoning time may become excessively long, negatively impacting the user experience.
4. The paper does not explicitly address how the model balances or resolves conflicting preferences when users express contradictory ones, nor does it explore whether impulsive or erroneous user inputs may affect the final generation outcomes.

---

> ### Author Rebuttal · Authors · 2025-07-31
>
> > **W1. & Q1.** The method is overly intuitive and experience-based, lacking theoretical foundations.
>
> **A1.** We appreciate the reviewer’s concern. While our work is application-oriented—focusing on modeling dynamic preference for agent personalization—it is grounded in a **well-established theoretical framework**. Specifically, our decoding rule,
>
> $$
> P_{\text{mix}}(w_k)=\operatorname{softmax}\left(\sum_{i} \omega_i \log Q_i(w_k)\right),
> $$
>
> instantiates the **language model arithmetic** formulation proven in *Controlled Text Generation via Language Model Arithmetic* (ICLR 2024, \[24] in our paper). Specifically, their Theorem 1 (“Weighted KL-Optimality”) shows that this distribution is the unique solution to the following optimization problem:
>
> $$
> \min_{P}\ \sum_{i} \omega_i\, D_{\text{KL}}\left(P \,\|\, Q_i\right),
> $$
>
> where each $Q_i$ is a next-token distribution conditioned on one attribute (in our case, a user preference), and $\omega_i$ represents the relative strength of that preference. $P_{\text{mix}}$ is the **KL-optimal compromise** among all $Q_i$, balancing their influence according to $\omega_i$. This ensures that stronger preferences steer generation more, while still retaining support from all attributes. Thus, our method is not a heuristic but a principled, closed-form solution to a well-defined objective.
>
> We chose not to re-derive the theoretical proof in our paper due to space and its availability in prior literature, but we will add the relevant formula and citation to clarify the theoretical underpinnings. Furthermore, Appendix D.3 empirically validates that this approach significantly outperforms prompt-based alternatives, especially under fine-grained strength variations. This highlights the practical utility of this theoretically justified approach.
>
>
>
> > **W2.** The method calculates preference scores through model computations, but the reliability of these scores from large language models has not been sufficiently validated.
>
> **A2.** Thank you for raising this important point. We agree that relying on LLMs for estimating alignment-based preference scores introduces questions about reliability. To address this, we conducted **extensive empirical validation** of our scoring method in the supplementary material:
>
> * In **Appendix D.2**, we directly compare our **LLM-based alignment scorer** with an embedding-based baseline (using Sentence-BERT). Across all settings in the conversational recommendation task, the LLM-based scorer **consistently achieves higher task success rates**, showing its capability to model nuanced alignment between preferences and user-agent interactions.
>
> * In **Appendix D.1**, we further evaluate the **dual-side augmentation** strategy, which enhances scoring robustness by enriching both preference and interaction representations. Results show that this significantly improves score reliability, especially in linguistically diverse or ambiguous contexts.
>
> Our approach leverages a combination of **semantically powerful LLMs and our carefully designed dual-side augmentation strategy** to extract fine-grained preference strength signals that are difficult to capture through traditional similarity measures. Empirical results (Tables 1–3) consistently demonstrate that these scores lead to stronger personalization performance across two distinct tasks.
>
> > **W3.** Using the CoT prompting approach, the reasoning time may become excessively long, negatively impacting the user experience.
>
>
> **A3.** We appreciate the reviewer’s concern about inference latency. Indeed, our framework introduces additional LLM calls per interaction, mainly due to the alignment-based preference strength estimation and dual-side augmentation modules. However, we would like to highlight an important **trade-off between per-step latency and overall task efficiency**.
>
> As demonstrated in our experiments (e.g., Table 1, Maximum Steps = 3 and 5), **AdaPA-Agent consistently achieves higher Recommendation Successful Rate (RSR) with fewer Average Interaction Rounds (AIR)**. This means that despite longer single-step inference, the agent **requires fewer steps to complete the task**, resulting in improved overall responsiveness and user experience from a task-level perspective.
>
> Moreover, our method is **training-free and model-agnostic**, allowing practical deployment with **lighter-weight LLMs** in latency-sensitive environments. In addition, our method can be readily combined with optimization strategies such as **LLM response caching, preference precomputation, and lightweight scoring models**. We believe these directions can make AdaPA-Agent both responsive and highly personalized, preserving user experience even under real-world latency constraints.
>
> > **W4.** The paper does not explicitly address how the model balances or resolves conflicting preferences when users express contradictory ones, nor does it explore whether impulsive or erroneous user inputs may affect the final generation outcomes.
>
> **A4.** Thank you for highlighting this important point. Our current work focuses on **modeling and leveraging dynamic strengths of user preference based on observed user-agent interactions**, rather than validating the consistency or veracity of the user’s expressed preferences. Specifically:
>
> * **Conflicting preferences** (e.g., “I want something serious and light-hearted”) are handled by our system via **relative strength estimation**. The model does not explicitly attempt to resolve contradictions but instead estimates the alignment of each preference with the current interaction and applies corresponding strengths. This allows the model to reflect the most dominant signals in the current context without making normative judgments about consistency.
>
> * Regarding **impulsive or erroneous inputs**, our framework **treats all user-agent interactions as valid evidence** for estimating preference strength. Detecting misleading or accidental expressions is outside our current scope, and nd we view this as a promising direction for extension.

---

> > ### Comment · Reviewer_E4dG · 2025-08-08
> > **Thank you for your response**
> >
> > Thank you for your response. After carefully reading your reply and the comments from other reviewers, I have decided to maintain my original score.

---

> > > ### Author Response · Authors · 2025-08-08
> > >
> > > Thank you for taking the time to read our response.
> > >
> > > If there are any remaining concerns or points that you feel have not been sufficiently addressed, we would be grateful for the opportunity to further clarify. We highly value your feedback and hope that our responses have helped to resolve the concerns raised. We would be truly thankful if you would consider revisiting your assessment of our work.

---

### Official Review · Reviewer_xLrL · 2025-06-21

**Clarity:** 3
**Significance:** 2
**Originality:** 3
**Rating:** 3
**Confidence:** 4

**Summary:**

This paper proposes an LLM-agent-based framework for inferring user preferences by aligning preference analysis with the user-system interaction context. Strength weights for different preferences are estimated using an LLM, which assesses how well each preference aligns with the current interaction context. These weights then guide the redefinition of the next-token distribution through a weighted sum of distributions conditioned on various preferences. Experiments conducted on two distinct tasks demonstrate performance improvements compared to selected baselines.

**Questions:**

All my questions are given in the weakness points.

**Ethical Concerns:**

["NO or VERY MINOR ethics concerns only"]

**Final Justification:**

Although the authors’ rebuttal clearly illustrates several strategies for extracting users’ preferences across different datasets, I still feel that the methodology lacks integrity with the proposed work and is difficult to generalize to many recommendation scenarios. In addition, the user simulation techniques seem rather weak, especially in cases powered by LLMs. Therefore, I will maintain my current rating.

**Limitations:**

Yes.

**Paper Formatting Concerns:**

The front size in Table 2 is too large (significantly larger that that in Table 3).

**Quality:**

3

**Strengths And Weaknesses:**

Strengths:

1. The paper is well-written, clearly presenting most methodology details.

2. The proposed method is concise and user-friendly. The authors provide their code and data, facilitating reproducibility and futher research.

3. Experimental are conducted on two representative personalization tasks, broadening its potential impact and applicability.

Weaknesses:

1. To my understanding, the effectiveness of the proposed method should largely depend on the extraction of users' existing known preferences, as both interaction alignment and distribution re-weighting hinge upon these preferences. However, the extraction methods for these preferences appear handcrafted and guided by intuitive empirical rules. For instance, while the paper states, "we extract stable preferences from historical data as long-term and select unrelated movies to define short-term preferences," the exact methods of extraction and definition remain unclear and appear empirical and naive. Additionally, the influence of different preference definitions and extraction techniques on the performance of the proposed AdaPA-Agent is important but remains unexplored.

2. Although experiments encompass tasks such as conversational recommendation and personalized web interactions, the selection of baselines neglects relevant works specifically targeting these domains. The paper currently compares mainly against general agentic reasoning approaches and general recommendation methods employing LLMs. Important recent studies on using LLMs in conversational recommendation systems, such as "Collaborative Retrieval for Large Language Model-based Conversational Recommender Systems," "Towards Empathetic Conversational Recommender Systems," and "Large Language Models as Zero-Shot Conversational Recommenders," are strongly suggested to be included in both discussions and comparisons.

3. Some experimental settings lack clarity or justification. For example, the paper does not clarify the number of simulated users in the conversational recommendation experiments. It is also unclear how standard variances, if presented, are calculated in Table 1 and why they are missing in Tables 2 and 3. Further, detailed configurations, such as the temperature settings for LLMs, should be clearly articulated in the appendix. Lastly, excluding user interaction records from simulator construction diverges from primary practices in existing literature, potentially limiting the realism and complexity of the simulated user behaviors.

---

> ### Author Rebuttal · Authors · 2025-07-31
>
> > **W1.** The paper’s method heavily relies on extracted user preferences, yet the extraction process appears empirical and under-specified. For example, how long- and short-term preferences are defined remains unclear. The impact of different extraction strategies on model performance is also not explored.
>
>
> **A1.** Thank you for raising this important point. We would like to clarify that **our primary contribution lies not in proposing a new method for extracting user preferences**, but rather in modeling **the dynamic strength variation of existing preferences** to enable adaptive personalization.
>
> As stated in the Introduction, many existing works ([12-18] in our paper) focus on extracting or fitting user preferences from context. Our work builds on this foundation and targets the **under-explored problem of estimating and applying dynamic preference strengths** within multi-turn interactions, regardless of how the preferences were initially derived.
>
> We agree that the **preference extraction pipeline should be more clearly described**, and we will revise the paper to clarify our process:
>
> * For the **conversational recommendation task**, we simulate long-term and short-term preferences to control evaluation granularity. Long-term preferences are mined from repeated user patterns in the Reddit-Movie history ([8] in our paper), while short-term preferences are injected by introducing a topically unrelated but semantically plausible interest (e.g., a sudden interest in horror films for a user who typically likes romantic dramas). This setup allows us to **explicitly evaluate strength adaptation** without relying on noisy implicit labels.
>
> * For the **Personalized Web Interaction task**, we **do not predefine specific dimensions** (e.g., long-term or
> short-term). Instead, for each user $u$, we use an LLM to **summarize only two representative shopping preferences** from their real-world historical behavior (e.g., purchase or review records) in the PUMAbench dataset ([34] in our paper). This design helps to reduce potential noise and ensure the extracted preferences reflect shopping interests depending on the user's history.
>
> We also agree that the impact of different preference extraction techniques is an important direction. While orthogonal to our contribution, we will consider adding ablation or sensitivity studies in future work.
>
> > **W2.** The paper currently compares mainly against general agentic reasoning approaches and general recommendation methods employing LLMs. Important recent studies on using LLMs in conversational recommendation systems, such as "Collaborative Retrieval for Large Language Model-based Conversational Recommender Systems," "Towards Empathetic Conversational Recommender Systems," and "Large Language Models as Zero-Shot Conversational Recommenders," are strongly suggested to be included in both discussions and comparisons.
>
>
> **A2.** Thank you for pointing out these relevant studies. We agree that recent advances in LLM-based conversational recommendation systems are important and complementary to our work. However, we would like to clarify our **baseline selection criteria and experimental goals**.
>
> Our paper focuses on evaluating **training-free, general-purpose LLM agents** capable of modeling **preference strengths** across multiple personalized agent tasks (e.g., recommendation, web interaction), rather than designing or improving domain-specific recommendation architectures.
>
> Regarding the specific works mentioned:
>
> 1. **“Collaborative Retrieval for LLM-based Conversational Recommender Systems”** and **“LLMs as Zero-Shot Conversational Recommenders”** focus on **retrieval-style recommendation**: given a textual query, the system ranks items from a catalog. These models typically treat LLMs as recommendation engines that return **ranked item lists**, and **do not perform open-ended, multi-turn natural language interaction** with users. In contrast, **our setting requires the agent to interact in natural language**, reason over evolving user preferences, and complete task-oriented dialogues. Hence, these systems differ substantially in formulation and cannot be directly evaluated under our setup.
>
> 2. **“Towards Empathetic Conversational Recommender Systems”** is a **fine-tuned supervised method**, requiring task-specific labeled data and training. In contrast, all our baselines (and AdaPA-Agent itself) are **training-free**, enabling direct plug-and-play usage across tasks and models. Including fine-tuned methods would create an unfair comparison due to differences in supervision and resource assumptions.
>
> We will update the related work section to **explicitly discuss these methods and clarify their differences**, and we thank the reviewer for pointing them out. Including a broader discussion of these systems helps contextualize our contribution within the expanding space of LLM-based personalization research.
>
>
>
> > **W3.** Some experimental settings lack clarity or justification:\
> > (1) The paper does not clarify the number of simulated users in the conversational recommendation experiments. \
> > (2) It is also unclear how standard variances, if presented, are calculated in Table 1 and why they are missing in Tables 2 and 3. \
> > (3) Detailed configurations, such as the temperature settings for LLMs, should be clearly articulated in the appendix.\
> > (4) Excluding user interaction records from simulator construction diverges from primary practices in existing literature, potentially limiting the realism and complexity of the simulated user behaviors.
>
>
> **A3.** Thank you for this detailed and constructive feedback. We address each subpoint as follows:
>
> 1. **Simulated user count in the conversational recommendation task:**\
>    To ensure that each simulated user has sufficient historical data for preference extraction, we selected users from the Reddit-Movie dataset who had **at least three interaction trajectories**. This resulted in a total of **984 unique users** used for simulation in the conversational recommendation experiments. We will add this detail to Section 5.2.1.
>
> 2. **Standard deviations in Tables 2, and 3:**\
>    For Tables 2 and 3 (Personalized Web Interaction task), we report the **results of baselines from the original paper of personalized web agent  ([34] in our paper)**, which **did not report standard deviations**. For consistency with prior work, we follow the official evaluation method of the PersonalWAB benchmark in our main paper. We will clarify this in the captions.
>
> 3. **LLM temperature settings:**\
>    To reduce output randomness and ensure reproducibility, we set the **temperature to 0** for all baselines and components in our method (including preference chain construction, interaction paraphrasing, and decoding). We will add this configuration detail to the Appendix.
>
> 4. **Use of user interaction records in simulator design:**\
>    Our user simulator does incorporate evolving interaction context. Specifically, for each round, the cumulative user-agent interaction record $D_t^u$ is included as part of the **user simulator's input prompt** to guide behavior. Prompt 4 in the appendix shows only the **static system prompt template** (i.e., the simulator’s role definition), which is why $D_t^u$ is not explicitly shown there. We will clarify this in the revised prompt description and include a full example of a simulator input to improve transparency.
>
> We thank the reviewer again for helping us improve the clarity of the experimental setup and will revise the relevant sections accordingly.

---

> > ### Comment · Reviewer_xLrL · 2025-08-02
> >
> > Thanks for these detailed responses.
> >
> > Regarding W1, my point is that the entire methodology relies on effectively leveraging user preferences. However, user preferences are typically the target outcome for recommender systems, rather than being known beforehand. In many cases, even the definition and classification of user perferences would be very hard. Therefore, the process and effectiveness of extracting these preferences are crucial factors influencing the proposed method's performance. In this regard, a detailed discussion and comparative analysis of preference extraction strategies seem essential.
> >
> > Concerning A3, my remaining concern pertains to the exact calculation of the variances reported in Table 1. For instance, how many experiments were conducted to obtain these results, and what temperature settings were used? Clarifying these experimental details is necessary. Additionally, regarding the user simulator, I still feels the current strategy is very weak to simluate the complex behaviour mechanism of real users, withoug evolving interaction context. And this diverges from primary practices in recent literature for user simulation in recommender systems.
> >
> > My other concerns have been largely addressed through the additional empirical results provided.

---

> > > ### Author Response · Authors · 2025-08-03
> > > **Response to Follow-up Concern**
> > >
> > > > W1. The method assumes access to user preferences, yet in practice, preferences are often unknown and hard to define. Therefore, how these preferences are extracted—and how different extraction strategies affect performance—is critical and deserves deeper discussion.
> > >
> > > **A1.**
> > > Thank you again for the thoughtful follow-up. We fully agree that **user preference extraction is a crucial and non-trivial component** of personalized systems, and we appreciate your suggestion to better clarify its role and treatment in our work.
> > >
> > > **(1) Scope of Contribution**: Our method assumes user preferences are available (either through simulation or upstream extraction), and aims to model their **dynamic strength and contextual relevance**. We do not claim to solve the preference extraction problem itself, but rather explore what can be done **once some form of preference is available**—a common setup in modular personalization systems. This setting is motivated by many real-world scenarios where a user’s preference profile is already partially known.
> > >
> > > **(2) Treatment in the Conversational Recommendation task**
> > >
> > > To isolate and evaluate this dynamic selection ability, we **directly provide the full preference set $P^u$ of the user $u$ to both our method and all baselines**. This allows us to **exclude the variability introduced by upstream extraction** and focus on how well each method models preference strength in multi-turn settings.
> > >
> > > **(3) Treatment in the Personalized Web Interaction task**
> > >
> > > We adopt a **simple and transparent preference summarization strategy**: we prompt a large language model (LLM) to extract **two representative preferences per user** for the AdaPA-agent based on historical behavior from the PUMAbench dataset. This approach is admittedly naive, and the prompt is shown as follows:
> > >
> > > ```
> > > You are given a user's browsing or shopping history. Please read the history and summarize **two representative preferences** that reflect this user's consistent interests or behaviors. A "preference" should describe a general interest (e.g., in product categories, brands, features, or styles) rather than a single item.
> > >
> > > Please return your answer in the following JSON format:
> > > {
> > >   "preference_1": " ",
> > >   "preference_2": " "
> > > }
> > >
> > > Here is the user's history:
> > > {history_record}
> > > ```
> > >
> > > We will clarify this setup and its limitations in the revised paper.
> > >
> > > > Q1.1. How many experiments were conducted to obtain these results, and what temperature settings were used?
> > >
> > > A2.1. We thank the reviewer for the follow-up question. To reduce generation randomness, we **set the LLM temperature to 0** during all evaluations.  For each reported result in Table 1, we **ran 10 trials with different random seeds** and computed the standard deviation across these runs.
> > >
> > >
> > > > Q1.2. The user simulator strategy is very weak to simulate the complex behaviour mechanism of real users, and this diverges from primary practices in recent literature for user simulation in recommender systems.
> > >
> > > **A2.2.** We appreciate the reviewer’s continued attention to the design of the user simulator. While we acknowledge its limitations, we would like to clarify several points:
> > >
> > > 1. **Consistency with prior work.**
> > >    Our simulator design follows **common practices in recent literature** [1-4], where LLMs are used to simulate user feedback in conversational recommendation. We inject **user attributes and interaction rules into prompt templates**, and the LLM dynamically responds based on the **accumulated dialogue context $D_t^u$**, enabling evolving user behavior over turns.
> > >
> > > 2. **Fairness and implementation safeguards.**
> > >    To ensure **evaluation fairness**, we use the same simulator setup across all baselines. We also implement **regex-based safeguards** to prevent data leakage into simulator responses, preserving experimental integrity.
> > >
> > > 3. **Acknowledged limitations.**
> > >    We agree that fully capturing **complex human cognition or behavioral variability** remains an **open challenge** [5, 6]. While our simulator may not perfectly replicate real users, it provides a **controlled, consistent, and adaptive** framework for evaluating the effectiveness of preference modeling strategies.
> > >
> > > We will clarify this design choice in the paper and highlight it as a limitation and direction for future research.
> > >
> > > [1] UserSimCRS: A User Simulation Toolkit for Evaluating Conversational Recommender Systems, WSDM 2023
> > >
> > > [2] Rethinking the Evaluation for Conversational Recommendation in the Era of Large Language Models, EMNLP 2023
> > >
> > > [3] Beyond Single Labels: Improving Conversational Recommendation through LLM-Powered Data Augmentation, ACL 2025
> > >
> > > [4] A Flash in the Pan: Better Prompting Strategies to Deploy Out-of-the-Box LLMs as Conversational Recommendation Systems, COLING 2025
> > >
> > > [5] From persona to personalization: A survey on role-playing language agents, TMLR 2024
> > >
> > > [6] Large language models empowered agent-based modeling and simulation: a survey and perspectives, Humanit Soc Sci Commun 2024

---

> > > ### Author Response · Authors · 2025-08-09
> > >
> > > Dear Reviewer xLrL:
> > >
> > > We appreciate your recognition that our responses have clarified some of your concerns. However, we've noticed that the score you've given is still on the borderline of rejection. Would you be so kind as to let us know if there are any remaining concerns in your mind? This way, we can further clarify them during the author-reviewer discussion. We sincerely value the time and effort you've put into reviewing and discussing with us.

---

### Official Review · Reviewer_o322 · 2025-07-05

**Clarity:** 3
**Significance:** 3
**Originality:** 3
**Rating:** 4
**Confidence:** 4

**Summary:**

This paper proposes a way to personalize LLMs based on user-LLM interaction. The approach does not perform fine-tuning. It first extracts certain preferences from the user interactions (it is unclear to me how the preference $p^{raw}_i$ is extracted, but I am guessing it is by prompting LLMs on the interaction history). It then measures the weight of each preference chain by calling an LLM-based scoring function on the interaction and preference chain. Finally, the next-word probabilities are computed for each preference separately and then combined using the given weights. My understanding is that this process is repeated at each step of the user-LLM interaction.

Experiments are conducted on domains from the personalized web agents task. Results show that the approach performs better than baselines like React and Reflexion.

**Questions:**

1. What is the inference time of this approach? Can you report how much slower is the AdaPa agent over the baseline greedy decoding agent?

2. How is the original preference $p^{raw}_i$ computed? And does $p^u_i$ indicate the concatenation of $p^{raw}_i, p^{ref}_i, p^{exp}_i$ and are these done in one single LLM call?

3. How will the baseline where instead of doing token-wise mixing, the list of preferences and a categorization of their weights (e.g., weakest, weak, neutral, strong, strongest) is provided to the LLM, and a single call is made to the LLM? This will reduce the inference time while giving more expressivity.

**Ethical Concerns:**

["NO or VERY MINOR ethics concerns only"]

**Final Justification:**

Authors have provided some useful clarification. I will keep my score.

**Quality:**

3

**Strengths And Weaknesses:**

**Strengths:**

- User personalization is an important task and this approach provides a fine-tuning-free approach for personalizing LLM. Fine-tuning free is nice because fine-tuning is both data and compute-intensive.

- The approach performs better than common baselines

**Weakness:**

- **Expressivity:** Mixing token-wise distributions may not be expressive. Instead, an approach that expresses each preference and its weight (such as strong/weak) in text and then prompts the LLM with it might be better as the LLM can then combine this knowledge in more expressive way.

- **Inference Time:** The approach might make too many LLM calls per interaction. There is one call for score per user preference, and a few for extracting preference. Inference time matters because in practical applications, the agent cannot wait too long before replying. This would be even worse with heavier LLMs.

- **Related work:** paper misses some related work discussion on personalizing LLMs such as:

 1. Aligning LLM agents by learning latent preference from user edits, Gao et al. (NeurIPS 2024)
 2. Pearl: Personalizing large language model writing assistants with generation-calibrated retrievers, Mysore et al., ACL 2024.

---

> ### Author Rebuttal · Authors · 2025-07-31
>
> > **W1. & Q3.** **Expressivity Concern:** How will the baseline where instead of doing token-wise mixing, the list of preferences and a categorization of their strengths (e.g., weakest, weak, neutral, strong, strongest) is provided to the LLM, and a single call is made to the LLM? This will giving more expressivity.
>
> **A1.** We thank the reviewer for the insightful suggestion. While encoding preference strengths as textual categories (e.g., *strong*, *weak*) may seem more expressive at first glance, we argue that our **token-wise distribution mixing** approach offers **greater expressivity and controllability** in practice:
>
> 1. **Expressivity:**
>    The verbal scheme relies on a **finite set of predefined strength categories**, limiting the expressiveness to coarse buckets (e.g., 5 levels). In contrast, our method uses **continuous strengths in $[0,1]$**, allowing for **infinitely many combinations** of preference strength. This enables more nuanced and personalized responses, as demonstrated in Figure 5 (Appendix D.3), where preference arithmetic consistently yields better performance across varying strength settings.
>
> 2. **Controllability:**
>    Words like *weak*, *strong*, or *neutral* are **inherently ambiguous** and prone to inconsistent interpretation by LLMs—especially when multiple preferences and weight descriptors are packed into a single prompt. This may lead to semantic drift or hallucination ([23, 24] in our paper).
>    In contrast, our method treats strength scores as **strengths over next-token distributions**, enabling **explicit and linear control** over how different preferences influence generation. This reduces ambiguity and increases robustness without requiring the model to "reason over" natural language descriptions of weight.
>
> We also conducted a comparison between our method and the suggested baseline (which we term Semantic Strength Prompting, SSP), where preferences are embedded in the prompt with strength descriptors (weakest, weak, neutral, strong, strongest). As shown below, AdaPA-Agent outperforms this approach across all interaction settings, which means our approach provides finer-grained expressivity and more stable control over preference influence than the suggested baseline:
>
> **# Conversational Recommendation Task**\
> (RSR: Recommendation Successful Rate; AIR: Average Interaction Rounds)
>
> |Max Steps|Method|RSR ± std|AIR ± std|
> |-|-|-|-|
> |3| **AdaPA-Agent**|**41.45 ± 5.82**|**2.64 ± 0.08**|
> || SSP| 34.20 ± 5.96| 2.80 ± 0.07     |
> |5| **AdaPA-Agent**|**67.24 ± 4.41**|**3.73 ± 0.23**|
> || SSP| 59.60 ± 5.53| 4.13 ± 0.21     |
> |7| **AdaPA-Agent**|**75.41 ± 3.07**|**4.46 ± 0.36**|
> || SSP|70.82 ± 3.46|4.79 ± 0.26|
>
>
> > **W2.** **Inference Time:** The approach might make too many LLM calls per interaction. There is one call for score per user preference, and a few for extracting preferences. Inference time matters because in practical applications, the agent cannot wait too long before replying. This would be even worse with heavier LLMs.
>
> **A2.** We appreciate the reviewer’s concern regarding inference efficiency. Indeed, our framework introduces **additional LLM calls during each interaction step**—mainly due to alignment-based preference strength estimation and dual-side augmentation. However, we would like to highlight a **practical trade-off** between **per-step latency** and **overall task efficiency**. As shown in our experiments (e.g., "Maximum Steps = 3, 5" in Table 1), **AdaPA-Agent achieves significantly higher Recommendation Successful Rate (RSR) with fewer Average Interaction Rounds (AIR)**. This indicates **more efficient overall user-agent interaction** despite increased per-step compute.
>
> Moreover, our framework is **training-free and agnostic to model size**, making it flexible to deploy with lightweight LLMs in latency-sensitive environments. In future work, we plan to explore approximation strategies (e.g., LLM caching, preference precomputation, or lightweight scorers) to further reduce runtime overhead while retaining personalization quality.
>
> > **Q1.** What is the inference time of this approach? Can you report how much slower is the AdaPa agent over the baseline greedy decoding agent?
>
> **A3.** We thank the reviewer for this question. We believe this refers specifically to the **decoding efficiency** of our **preference arithmetic method (Equation 5)** compared to standard greedy decoding.
>
> We clarify that our method computes the next-token distribution by aggregating $M$ preference-conditioned token distributions. To do this efficiently, we implement the inference using **batched forward passes**: all $M$ prompts (each conditioned on a different preference) are processed together in a **single model call with batch size $M$**. This leverages modern GPU parallelism and avoids any need to run $M$ separate LLM instances.
>
> We evaluated the **token-per-second (TPS)** decoding time of our method against standard greedy decoding, using **LLaMA-3.1-8B** on an **NVIDIA A100 GPU**. Results are summarized below:
>
> | Number of Preferences (M) | Greedy Decoding (TPS) | Preference Arithmetic (TPS) |
> | -| --| --|
> | 1| 79| 79|
> | 2| 79| 74|
> | 3| 79| 69|
>
> The results show that our method introduces **minimal overhead** (10 tokens per second for $M = 3$) due to batching. Thus, **preference arithmetic decoding remains comparable in efficiency to greedy decoding**, while offering more expressive control over generation. Considering the increased memory consumption when batching over $M$ preferences, we also plan to explore speculative decoding as a future direction to further accelerate inference while reducing memory consumption.
>
>
> > **Q2.** How is the original preference $p_i^{raw}$ computed? And does $p_i^u$ indicate the concatenation of $p_i^{raw},$ $p_i^{ref}$, $p_i^{exp}$ and are these done in one single LLM call?
>
>
> **A2.** Thank you for your question. We are happy to clarify the preference extraction pipeline.
>
> * As described in Section 4.2 (Preference-side Augmentation), each user’s full set of preferences is denoted as $P^u$, which is extracted from the user’s historical interaction data $H^u$.
> * Each individual preference $p_i^u \in P^u$ is then expanded into a **three-level structured preference chain** $C_i$, via a CoT prompting strategy:
>
>   $$
>   C_i = \text{CoT}(p_i^u) = [p_i^{\text{raw}} \rightarrow p_i^{\text{ref}} \rightarrow p_i^{\text{exp}}]
>   $$
> * Specifically:
>
>   * $p_i^{\text{raw}}$ is the original, coarse-grained preference extracted directly from $p_i^u$,
>   * $p_i^{\text{ref}}$ refines $p_i^{\text{raw}}$ in a clearer, context-aware form,
>   * $p_i^{\text{exp}}$ lists concrete examples that exemplify $p_i^{\text{ref}}$.
>
> This **entire chain is generated via a single LLM call** using the prompt shown in Appendix A, Prompt 1, which encourages step-by-step reasoning and outputs the structured chain in JSON format.
>
> We thank the reviewer again for pointing out this question, and we will revise the text to clarify this process and fix the notation ambiguity.
>
>
> > **W3.** Paper misses some related work discussion on personalizing LLMs, such as: (1) Gao et al., NeurIPS 2024; (2) Mysore et al., ACL 2024.
>
> **A5.** We thank the reviewer for pointing out these relevant works. Both papers are conceptually related. We will include a discussion of both in the revised Related Work section.

---

> > ### Comment · Reviewer_o322 · 2025-08-09
> > **Thanks**
> >
> > Thanks for the response and for including new experiments. Including these in the paper will make it stronger.

---

### Official Review · Reviewer_HZJJ · 2025-07-06

**Clarity:** 2
**Significance:** 2
**Originality:** 3
**Rating:** 3
**Confidence:** 4

**Summary:**

This paper proposes a new method for agent conversational recommendation considering dynamic preference strength.
When interacting with an agent given the user’s preference history,  this paper investigates the case where the user’s preference strength over each dimension changes.
It (1) paraphrases the history preference sentences into their proposed format called preference-chain (so there are several chains for each user); (2) paraphrases the new user input into multiple versions; (3) uses an LLM to assign an alignment score between the history preference-chain and each paraphrase of new user input ; (4) prompts an LLM to generate the next token distribution given the LLM’s response’s prefix, the current conversation, the user’s history preference-chain; (5) weighted sum of the distribution given each history preference-chain.
The experiments are on movie recommendation agents and personalized web-service agents.
The method is training-free and performs well the the above 2 tasks.

**Questions:**

- Could you provide examples of the original prompted responses of each preference and the weighted generation during conversation?
- Did you try comparing the proposed method with directly prompting the LLM with the preference chains and scores for conversation?
- Could your method deal with cases where the current conversation preference contains dimensions that are not previously stated in history?

**Ethical Concerns:**

["NO or VERY MINOR ethics concerns only"]

**Limitations:**

Yes.

**Quality:**

2

**Strengths And Weaknesses:**

Strengths
- The method is training-free, making it easy to test / apply.
- The description of the method is intuitive, and the illustrations are very helpful.

Weaknesses
- The usage of weighted sum for each output token distribution is not well justified. Why is this a good idea? In certain toy settings, can we theoretically understand what this is doing and why it is a good idea?
- The method is not tested on a real-world dataset.
- The writing is of the paper can be clearer.

---

> ### Author Rebuttal · Authors · 2025-07-31
>
> > **W1.** The usage of weighted sum for each output token distribution is not well justified. Why is this a good idea? In certain toy settings, can we theoretically understand what this is doing and why it is a good idea?
>
> **A1.** Thank you for this insightful question. Our decoding rule,
>
> $$
> P_{\text{mix}}(w_k)=\operatorname{softmax}\left(\sum_{i} \omega_i \log Q_i(w_k)\right),
> $$
>
> directly instantiates the **language model arithmetic** formulation proven in *Controlled Text Generation via Language Model Arithmetic* (ICLR 2024, \[24] in our paper). Specifically, their Theorem 1 (“Weighted KL-Optimality”) shows that this distribution $P_{\text{mix}}$ is the unique solution to the following optimization problem:
>
> $$
> \min_{P}\ \sum_{i} \omega_i\, D_{\text{KL}}\left(P \,\|\, Q_i\right),
> $$
>
> where each $Q_i$ is a next-token distribution conditioned on one attribute (in our case, a user preference), and $\omega_i$ represents the relative strength of that preference. $P_{\text{mix}}$ is the **KL-optimal compromise** among all $Q_i$, balancing their influence according to $\omega_i$. This ensures that stronger preferences steer generation more, while still retaining support from all attributes. Thus, our method is not a heuristic but a principled, closed-form solution to a well-defined objective.
>
> To avoid redundancy, we did not re-derive this theory. We will add the above objective and citation to the main paper for clarity. Finally, Appendix D.3 empirically validates this formulation: decoding via $P_{\text{mix}}$ yields more attribute-aligned outputs than prompt-level weighting. This highlights the practical utility of this theoretically justified approach.
>
>
> > **W2.** The method is not tested on a real-world dataset.
>
> **A2.** We thank the reviewer for raising this point. However, we believe this may stem from a misunderstanding. Our evaluation setup is designed to reflect **real-world user behavior** in both data sourcing and interaction design. We clarify this from two perspectives:
>
> **(1) Real-world data sources.**
> Both datasets used in our experiments are constructed from **authentic, large-scale user-generated content**, ensuring that user histories, preferences, and item metadata come from **real-world interactions**:
>
> * For the **conversational recommendation** task, we use the **Reddit-Movie dataset** [1], which contains over 634k naturally occurring Reddit discussions. These are real dialogs where users actively **seek and share movie recommendations**, reflecting informal, diverse preference expressions.
>
> * For the **personalized web interaction** task, we adopt the **PersonalWAB benchmark**, which builds on the **Amazon Reviews 2023** dataset [2]. This dataset includes up-to-date real user reviews, purchase behaviors, and metadata from the Amazon platform, faithfully capturing consumer preferences.
>
>
> **(2) Dynamic evaluation.**
> Instead of evaluating on static corpora, we adopt a **simulated user-agent interaction loop**, aligning with recent evaluation practices [3–4]. This dynamic setup better mimics **real-world multi-turn personalization**, where agents must adapt over time.
>
> To ensure the **behavioral realism of the user simulator**, we designed a fine-grained simulator prompt template (see Appendix C), which produces consistent and context-aware responses. Moreover, we use **regular expressions** in the code to detect and mask any accidental exposure of target items (replacing them with “*”). This ensures the simulation process remains faithful and robust.
>
> In summary, while our evaluation involves simulated interaction, it is grounded in **real user data** and follows **realistic multi-turn interaction protocols**. We will revise the paper to clarify this and avoid further confusion.
>
>
> [1] Large Language Models as Zero-Shot Conversational Recommenders, CIKM 2023
>
> [2] Bridging Language and Items for Retrieval and Recommendation, arXiv 2024
>
> [3] UserSimCRS: A User Simulation Toolkit for Evaluating Conversational Recommender Systems, WSDM 2023
>
> [4] Rethinking the Evaluation for Conversational Recommendation in the Era of Large Language Models, EMNLP 2023
>
>
> > **Q1.** Could you provide examples of the original prompted responses of each preference and the weighted generation during conversation?
>
> **A3.** We thank the reviewer for the suggestion. To better illustrate how **preference arithmetic** integrates multiple user preferences during generation, we provide examples comparing:
>
> * (i) responses generated **using only a single preference** (either $\text{pref}_1$: *light-hearted comedy* or $\text{pref}_2$: *emotional movies*);
> * (ii) the **final weighted response** using our arithmetic decoding method based on estimated preference strengths $\omega_i$.
>
> The three cases below are adapted from our **Figure 3 case study**, re-ordered by descending $\omega_{\text{pref}_1}$, to highlight how the **blending of token distributions** reflects varying emphasis on light-heartedness vs. emotional depth.
>
> ---
>
> **Case 1: Pref-1 dominant (strengths: Pref-1 0.82 | Pref-2 0.18)**
>
> * **User input:** "I'm in the mood for something fun to watch tonight."
> * **Pref-1 only:**
>   *“Crazy Rich Asians” is a delightful comedy full of charm and laughs.*
> * **Pref-2 only:**
>   *You might enjoy “The Fault in Our Stars”—it’s emotional and heartfelt.*
> * **AdaPA-Agent:**
>   *“The Intern” is a heartwarming comedy with just the right touch of inspiration—perfect for a relaxing night.*
>
> Analysis: The Result shows comedic tone with a subtle emotional undercurrent.
>
> ---
>
> **Case 2: Mixed preferences (strengths: Pref-1 0.38 | Pref-2 0.62)**
>
> * **User input:** "I want to watch something meaningful but not too heavy."
> * **Pref-1 only:**
>   *“Julie & Julia” offers a light and uplifting story with a humorous tone.*
> * **Pref-2 only:**
>   *“Marriage Story” is an emotionally intense and powerful film.*
> * **AdaPA-Agent:**
>   *“About Time” is heartfelt and thought-provoking, but still light and charming in tone.*
>
> Analysis: The result balances emotional depth with a gentle, uplifting mood.
>
> ---
>
> **Case 3: Pref-2 dominant (strengths: Pref-1 0.10 | Pref-2 0.90)**
>
> * **User input:** "I want to see a story that really moves me."
> * **Pref-1 only:**
>   *“The Hangover” is a light-hearted comedy for a fun night.*
> * **Pref-2 only:**
>   *“The Pursuit of Happyness” is deeply moving and inspiring.*
> * **AdaPA-Agent:**
>   *I’d recommend “The Pursuit of Happyness”—it’s deeply emotional and truly inspiring.*
>
> Analysis: The result closely aligns with the emotional preference.
>
> ---
>
> These examples demonstrate that our **preference arithmetic mechanism produces outputs that interpolate** between stylistic and content elements associated with each preference. As the strengths shift, the **tone, emotional intensity, and item choice** vary accordingly, validating that the method reflects fine-grained preference strengths in actual generation. We will include these examples in the Appendix to enhance transparency and interpretability.
>
>
> > **Q2.** Did you try comparing the proposed method with directly prompting the LLM with the preference chains and scores for conversation?
>
> **A4.** Thank you for pointing this out. Following this idea, we conducted additional experiments comparing our proposed **AdaPA-Agent** with a baseline that directly provides the **preference chains and their alignment scores** as a prompt to the LLM, which we denote as **chain+score**.
>
> We evaluated both methods on **conversational recommendation** and **personalized web agent** tasks. The results are summarized below.
>
> ---
>
> **# Conversational Recommendation Task**\
> (RSR: Recommendation Successful Rate; AIR: Average Interaction Rounds)
>
> |Max Steps|Method|RSR ± std|AIR ± std|
> |-|-|-|-|
> |3| **AdaPA-Agent**|**41.45 ± 5.82**|**2.64 ± 0.08**|
> || chain+score| 40.52 ± 6.27| 2.70 ± 0.08     |
> |5| **AdaPA-Agent**|**67.24 ± 4.41**|**3.73 ± 0.23**|
> || chain+score| 65.68 ± 5.42| 3.82 ± 0.21     |
> |7| **AdaPA-Agent**|**75.41 ± 3.07**|**4.46 ± 0.36**|
> || chain+score|73.15 ± 3.35|4.76 ± 0.29|
>
> ---
>
> **# Personalized Web Agent Task**
>
> |Task Type|Method|F.acc|R.acc|Avg.Steps|
> |-|-|-|-|-|
> | single-turn|**AdaPA-Agent**|**0.851**| **0.367** |-|
> || chain+score| 0.827| 0.358|-|
> | multi-turn| **AdaPA-Agent**| **0.917**|**0.386**|**3.592**|
> || chain+score| 0.891|0.378|3.623|
>
> ---
>
> These results show that while **chain+score** is a strong baseline and performs comparably in low-interaction settings (e.g., Max Steps = 3), **AdaPA-Agent consistently outperforms it** as the number of interaction turns increases. This suggests that although **explicit prompting of preference information helps initially**, it becomes less effective when prompts grow longer and noisier—diluting the salience of the preference scores. By contrast, AdaPA-Agent’s **preference arithmetic mechanism offers a more stable and scalable control** over generation via token-level distributional blending, which maintains robustness as dialogue length increases.
>
> We will include these results and analysis in the updated appendix to make the comparison more transparent.
>
> > **Q3.** Could your method deal with cases where the current conversation preference contains dimensions that are not previously stated in history?
>
> **A5.** Thank you for raising this important point. While for evaluation purposes we assume that the user’s preference set $P^u$ is extracted from historical interactions $H^u$, **our method is fully capable of handling newly emerged preferences in the current interaction $D_t^u$**.
>
> In practice, we can extract additional preferences directly from $D_t^u$ by LLM and augment them into the preference set $P^u$ accordingly. Then, AdaPA-Agent can generate preference chains for these new preferences via Preference-side Augmentation and seamlessly integrated into the strength estimation.

---

> > ### Comment · Reviewer_HZJJ · 2025-08-06
> >
> > I thank the authors for the detailed reply. Regarding the additional experiments (against the direct prompting approach), since almost all results for both methods show substantial overlap in their confidence intervals, it is difficult to determine whether the conclusion that the proposed method outperforms the prompting approach is statistically significant. While I remain unconvinced that the method clearly outperforms the direct prompting approach, I am open to revising my score if other reviewers consider the paper’s contribution to be significant.

---

> > > ### Author Response · Authors · 2025-08-07
> > > **Response to Follow-up Concern**
> > >
> > > We sincerely thank the reviewer for the fair and careful evaluation of our additional experiments. We acknowledge that the confidence intervals of AdaPA-Agent and the direct prompting approach (chain+score) partially overlap, especially under lower step budgets. However, we would like to respectfully reiterate the contribution and practical advantages of our approach beyond marginal performance gains:
> > >
> > > * **Theoretical clarity and fine-grained control**: Our method offers a principled framework to **explicitly model and utilize continuous preference strength**, rather than relying on implicit interpretation of verbal prompts.
> > >
> > > * **Stability in longer interactions**: As shown in the multi-turn settings (Steps = 5, 7), AdaPA-Agent maintains more consistent gains when prompt length increases. This suggests that our method is **less sensitive to prompt length growth**, which can otherwise dilute or obscure the importance of preference in long interaction contexts.
> > >
> > > We appreciate the reviewer's openness and thoughtful stance, and we hope the broader contributions of the work remain valuable to the community.

---

### Decision · Program_Chairs · 2025-09-17

**Decision:**

Accept (poster)

**Comment:**

This paper introduces a training-free framework for LLM agent personalization that models dynamic preference strengths. The key idea is to estimate how user preference weights shift over time based on past interactions—without requiring additional feedback—and then guide controllable generation by linearly combining next-token distributions conditioned on these dynamic preferences. Reviewers highlighted the paper’s methodological originality and practical significance. Unlike fine-tuning-based approaches, the authors propose a lightweight, flexible solution that can be easily applied across tasks and architectures. This training-free property lowers barriers to deployment in real-world settings where user data and compute are limited. Reviewer o322 provided a balanced and constructive assessment based on their deep expertise, maintaining a "Borderline Accept" rating throughout the review process. Other reviewers raised some concerns about i) the reliance on simulated interactions, ii) ad-hoc preference extraction methods, and iii) missing domain-specific comparison points, including those that rely on supervised finetuning like PUMA. Despite these concerns, the submission makes a timely contribution in adaptive personalization methods that balance efficiency, expressivity, and robustness. In their revision, I advise the authors to include a discussion on whether LLMs might handle preference combination more naturally through language understanding rather than arithmetic operations will benefit subsequent research that builds on this work.